# Development of the oral resistome during the first decade of life

Smitha Sukumar [1] ✉, Fang Wang[1,2], Carra A. Simpson[3], Cali E. Willet [4], Tracy Chew [4], Toby E. Hughes [1,5], Michelle R. Bockmann[5], Rosemarie Sadsad [4], F. Elizabeth Martin[1], Henry W. Lydecker[4], Gina V. Browne[1,6], Kylie M. Davis[5], Minh Bui [7], Elena Martinez[1,8] & Christina J. Adler [1,2] ✉

Antibiotic overuse has promoted the spread of antimicrobial resistance (AMR) with significant health and economic consequences. Genome sequencing reveals the widespread presence of antimicrobial resistance genes (ARGs) in diverse microbial environments. Hence, surveillance of resistance reservoirs, like the rarely explored oral microbiome, is necessary to combat AMR. Here, we characterise the development of the paediatric oral resistome and investigate its role in dental caries in 221 twin children (124 females and 97 males) sampled at three time points over the first decade of life. From 530 oral metagenomes, we identify 309 ARGs, which significantly cluster by age, with host genetic effects detected from infancy onwards. Our results suggest potential mobilisation of ARGs increases with age as the AMR associated mobile genetic element, Tn*916* transposase was co-located with more species and ARGs in older children. We find a depletion of ARGs and species in dental caries compared to health. This trend reverses in restored teeth. Here we show the paediatric oral resistome is an inherent and dynamic component of the oral microbiome, with a potential role in transmission of AMR and dysbiosis.

The human microbiome includes reservoirs of antimicrobial resistance genes (ARGs) which are part of a network of resistomes (collections of ARGs) that exist in a variety of animal and environmental microbiomes. Surveillance of resistomes is a key tenet of the One Health approach to combating antimicrobial resistance (AMR). One Health[1] recognises the interconnectivity between human and animal health in our shared environment and is a transdisciplinary approach to tackling AMR, a worldwide health and economic issue. The oral microbiome comprises the second most significant microbial presence within the body (after the gut[2]) and is an important interface between the body and the external environment, e.g., eating[3]. Therefore, characterising the AMR potential of the human oral microbiome is important. Antimicrobial resistance genes have been identified in the oral microbiome throughout human history[4] and have been found in all ages of the human population from neonates to adults[5,6]. Moreover, the oral microbiome is a known site for acquisition and transfer of resistance via horizontal gene transfer (HGT), which can contribute to the systematic development of antimicrobial-resistant infections[7].

The acquisition and development of AMR amongst both commensal and pathogenic bacteria in the oral microbiome during childhood are largely unknown. Childhood is characterised by major changes in the composition of the oral microbiome, namely increasing

[1]Faculty of Medicine and Health, The University of Sydney, Sydney, NSW, Australia. [2]Charles Perkins Centre, The University of Sydney, Sydney, NSW, Australia. [3]The Vatche and Tamar Manoukian Division of Digestive Diseases, Department of Medicine, David Geffen School of Medicine, University of California, Los Angeles, Los Angeles, CA, US. [4]Sydney Informatics Hub, Core Research Facilities, The University of Sydney, Sydney, NSW, Australia. [5]Adelaide Dental School, University of Adelaide, Adelaide, SA, Australia. [6]Institute of Dental Research, Westmead Centre for Oral Health, Westmead, NSW, Australia. [7]Melbourne School of Population and Global Health, The University of Melbourne, Melbourne, Australia. [8]Institute of Clinical Pathology and Medical Research, NSW Health Pathology, Sydney, NSW, Australia. ✉e-mail: smitha.sukumar@sydney.edu.au; christina.adler@sydney.edu.au

diversity as children grow older[8]. These changes are driven by diet, with the introduction of solid foods, and the emergence of teeth, which provide a non-shedding surface for biofilm maturation[9,10]. While the environment is a known major modulator of the oral microbiome, host genetics[11] have also been shown to influence oral bacteria which carry ARGs. To examine the role of the environment in the development of the oral resistome, studies are required that are nested within a population structure designed to measure both the influence of the environment and host genetics. The classical twin study design allows comparison of a phenotype such as AMR between monozygotic (MZ) and dizygotic (DZ) twin pairs, to estimate host genetics in relation to the environment.

Tooth emergence in childhood also marks the potential for dental caries, which the Global Burden of Health Report (2016) found to be the most prevalent, microbially induced disease worldwide[12]. Dental caries are driven by environmental and host genetic factors where an imbalance in the plaque biofilm equilibrium leads to the demineralisation of tooth structure[13]. Treatment of advanced lesions relies on surgical intervention, usually restoring the cavity. Current restorative materials have zero to limited antimicrobial properties to modulate the biofilm to prevent further demineralisation. The development of bioactive dental restorative materials is highly desirable and the incorporation of new antimicrobial peptides like nisin into dental materials holds promise but may create resistance in oral microbiota[14]. Relationships between the dysbiotic oral microbiome and restorative materials with AMR are currently unexplored.

Here we show the widespread presence of ARGs in the oral microbiome, which significantly change in composition and potential interaction with other constituents of the microbiome over the first decade of life and in response to changes in oral health state. The results of this study present a longitudinal, genomic investigation of oral resistome development over the first decade of life (2.4 months – 10.8 years) using 530 oral metagenomes isolated from 221 Australian twins. Analysis of our large collection of oral metagenomes enabled investigation of three aims: (i) examination of the sequential development of the oral resistome including taxonomic and functional association, in addition to mobilisation potential of ARGs over the first decade of life; (ii) determination of the major environmental and host genetic contributions to the development of AMR using a twin study design and (iii) investigation of the contribution of AMR to the presence and severity of dental caries and response to placement of restorations.

## Results

### The oral resistome is characterised by compositional changes during dental development

Target read pairs after removal of human host DNA ranged from 11,215,924 to 89,678,704 pairs per sample, with a mean of 53,404,886 pairs (Supplementary Data 1, Fig. S1). The oral biofilm samples had a minimal level of host contamination, averaging 12% across all samples, compared to similar studies[15]. At time point 1 (T1. edentulous (no teeth)), we assessed 139 samples (average age $6.7 \pm 2.7$ months), 180 samples ($1.6 \pm 0.4$ years) at T2 (primary/deciduous/baby teeth only), and 211 samples at T3 (mixed dentition, $8.5 \pm 1.2$ years old). Of the children sampled, 55% were sampled at all three-time points, with only 12% of participants sampled at one-time point.

Genes associated with AMR were found in all 530 samples. Of the 437 genes identified, 128 were multidrug-resistant efflux pump (MDEP) genes which accounted for on average 35% at T1, 31% at T2, and 20% at T3 of the total relative abundance of the resistome. These genes were excluded from further analysis as they are not strictly defined as ARGs. The primary function of MDEP genes is detoxification rather than specifically conferring resistance[16]. The remaining 309 ARGs found in the childhood oral biofilm samples belonged to 23 AMR gene classes

(Fig. S2 and Supplementary Data 2). The number of AMR gene classes decreased over time (20 at T1, 23 at T2, and 17 at T3) with five classes (macrolides, β-lactams, macrolide/lincosamide, fluoroquinolones, and tetracycline) accounting for 90% of the total relative abundance across all time points. However, temporal variation was observed in certain classes e.g., β-lactams (15%, 25%, and 19% of total relative transcripts per million (TPM) at T1, T2, and T3 respectively) and tetracyclines 10% at T1 and 4% at T2 and T3 (Figs. 1, S2, Supplementary Information).

To investigate the influence of age on the oral resistome, we compared the abundance of ARGs across time points that mark childhood dental development. Principal component analysis (PCA) of normalised ARG abundance revealed significant clustering by the stage of dental development as indicated by time point (Fig. 2a, PERMA-NOVA, $p = 0.001$). Vector loadings from the PCA (Fig. S3) revealed infancy (T1) was discriminated by a higher abundance of tetracycline resistance genes, $tet(A)$ and $tet(B)$. Time point 2 (deciduous dentition) was typified by a higher abundance of β-lactamase resistance genes, $cfxA3$ and $penA$. In the mixed dentition (T3) there was increased abundance of macrolide resistance genes, $macA$ and $macB$. The ARGs identified from the PCA were found to be significantly associated ($p < 0.05$) with each of the identified time points by differential abundance analysis (DAA) (MaAsLin2[17], Table S1 Supplementary Information).

While the resistome shifted in composition with age, there was also a degree of consistency in the resistome across childhood. Throughout the first decade of life, 60 ARGs belonging to 15 AMR gene classes were present across all time points (Fig. 2b). The plurality of these ARGs were resistant to β-lactams (16 genes), however the most abundant AMR gene class within this subpopulation were macrolide resistant genes, accounting for 45% of the relative abundance of the resistome. Additionally, a core resistome (genes present at all time points with ≥95% prevalence[18]) - macrolide efflux genes, $mef(A)$ and $msr(D)$, and the macrolide/lincosamide resistance gene $RlmA(II)$ were identified.

### Resistome development mirrors the microbiome with increased mobilisation potential as children grow older

To gain an understanding of the AMR potential of the microbiome, we calculated the total number of identified ARGs/total number of genes. The resistome accounted for 0.10% at T1, 0.08% at T2 and 0.04% of the microbiome at T3 respectively. These percentages are lower than previously reported, however, the data was from an adult population analysed with a now decade old database[19].

We assessed whether the overall diversity of species and predicted functional pathways in the oral microbiome was predictive of the diversity of the resistome at each time point (Fig. 2c) using generalised estimating equation. Our results show that a higher α-diversity of species and functional pathways is predictive of higher resistome diversity values per point ($p <$ or $= 0.001$ at T1, T2, and T3) (Table S2, Supplementary Information).

To investigate the interaction between the abundance of species, predicted functional pathways, and ARGs, over time, we used a multi-block discriminant analysis approach (DIABLO[20], Fig. 3). Potential interactions between species and function accounted for most correlations (260), compared with four correlations found between the resistome and functional pathways. The resistome/function interactions involved two ARGs ($macA$ and $tetA(60)$). The tetracycline resistance genes were negatively correlated with nucleoside and nucleotide biosynthesis pathways (Supplementary Data 3). In comparison, $macA$ was positively correlated with mycothiol biosynthesis, with the highest abundances of these ARGs and pathways observed at T3 (Fig. S4).

As biofilms play a significant role in oral health and AMR (biofilms promote HGT between ARGs), we assessed if there was a correlation between resistome α-diversity (categorised as high (Shannon score >2.10) or low) and abundance of 66 predicted functional pathways

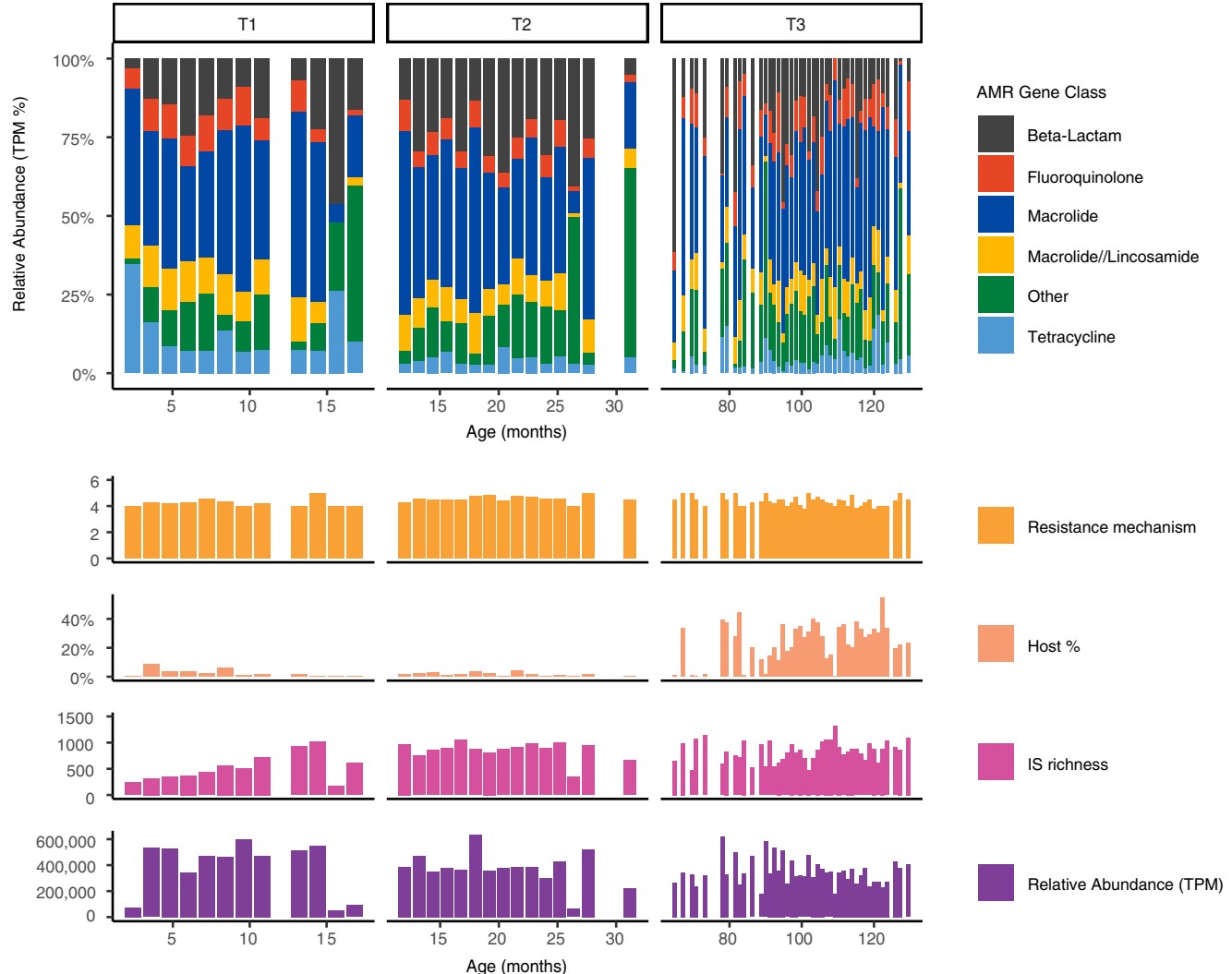

**Fig. 1 | Overall composition of the paediatric oral resistome over the first decade of life which is stratified by age and sampling time point (T1, T2, and T3).** The stacked bar chart shows the five most abundant AMR gene class types, normalised by transcripts per million (TPM) which accounted for over 90% of the resistome. Also displayed are the number of resistance mechanisms, amount of human contamination (host %), number of different types of insertion sequences (IS richness), and overall relative abundance of the resistome (TPM).

associated with biofilm promotion. We hypothesised that children with more diverse resistomes would have a higher abundance of biofilm-associated pathways. Six pathways were identified as significantly differentially abundant between the high and low resistome diversity groups (LEfSe[21]) and most (5/6) of these were associated with the high diversity group (Fig. S5). High diversity was associated with an increased abundance of pathways for sugar degradation and fatty acid biosynthesis. Low diversity was associated with an increase in abundance of the super pathway of pyrimidine nucleobases salvage, which enables salvaging of uracil for the generation of RNA.

The DIABLO correlation analysis revealed that overall, the resistome had more potential interactions with species than functional pathways. To further investigate this, we used ARG-carrying contigs of sufficient length[22] (median length = 11,538 bp across time), for correlation analysis between taxa and genes (Fig. S6). A total of 645 ARG-carrying taxa across the three time points were identified, with a total of 356 representative species selected to reconstruct a species tree using 16S rRNA sequences (Fig. 4 and Supplementary Data 4). The dominant genera in the childhood oral microbiome was *Streptococcus* spp, with *Streptococcus mitis*, commonly found in association with core ARGs, *mef(A) msr(D),* and *RlmA(II)*, at all three time points (Supplementary Data 5).

We identified the presence of transposable elements in the metagenomes, specifically insertion sequences (IS) to investigate the potential for ARG mobilisation within the oral microbiome. Insertion sequences were identified in all 530 metagenomes and the number of unique IS (richness) significantly increased between T1 and T3 (13.4% increase, *p* < 0.0001, paired t-test) (Table S2 Supplementary Information and Supplementary Data 6). Several IS families associated with ARG mobilisation were identified in our population (ISAs1, IS91, IS110, IS481, IS607, IS630, IS1182, and Tn1-22)[23] including the *Tn916* family of conjugative transposons which are implicated in oral ARG transfer[24]. While the *Tn916* transposases accounted for 1% of the total IS identified, they were highly prevalent across all time points (98% of samples). To better understand the mobilisation potential of the resistome, we identified contigs carrying both *Tn916* transposase and ARGs within 30kb[24] of each other (Supplementary Data 7). These had a mean distance across time points of 6394 bp. Co-carriage of *Tn916* and ARGs was observed in 50%, 32%, and 35% of individuals at T1, T2, and T3, respectively. Streptococcal species were associated with this co-carriage across all time points (78%). Over time *Tn916* transposase was found in more species and in combination with more ARGs. At T1 *Streptococcus oralis* had co-carriage of *Tn916* with *tet(M)* and *ermB*. By T3 the same species is associated with three additional ARGs *mefA, msrD,* and *lsa(C)* (Fig. 5).

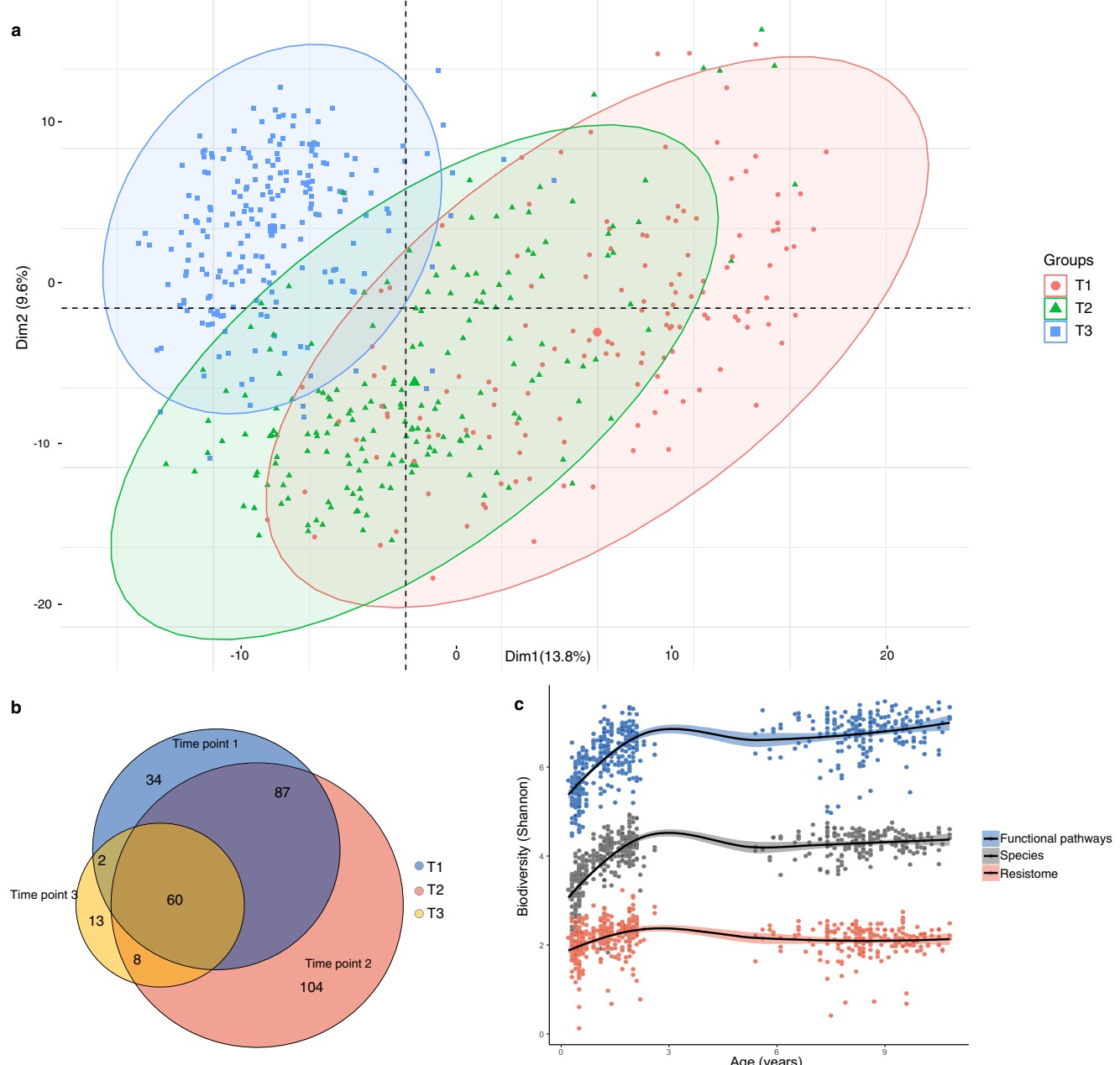

**Fig. 2 | Temporal changes in ARGs and the overall oral microbiome by dental development. a** Principal component analysis was performed on individuals using the TPM normalised ARGs that had been filtered to remove low abundance ARGs (below 0.01%), total sum scaled (TSS) and centre log ratio (CLR) transformed. Individuals are coloured by time point to observe temporal effects and 95% confidence intervals of groupings presented. **b** Distribution of unique and shared ARGs by time point. **c** α-diversity (Shannon Index) of the resistome, species makeup, and functional pathways present in the oral microbiome, plotted against age. Resistome diversity was calculated using TPM-normalised ARGs. Species and functional pathways diversity was calculated using counts per million (CPM) normalised abundance data generated from Kraken/Bracken and HUMAnN2, respectively. Shading per line represents the 95% confidence intervals for resistome, species, and functional diversity, respectively.

## Host genetic and environmental factors contribute to the composition of the developing oral resistome

We next investigated host-genetic and environmental associations with resistome composition at a global level (α-diversity) and ARG level. We specifically looked at the core ARGs (*mefA, msrD, RlmA(ll)*) and other genes identified as changing over time, including *macB, cfxA3, penA, patB,* and *tetA(46).*

We found an increasing phenotypic correlation between MZ twins, possibly reflecting greater host-genetic effects, based on their resistome diversity over time, rising from 15% at T1 to 49% at T2 and 52% at T3 compared to DZ twins (44% (T1), 28% (T2) and 14% (T3)).

Using a variance components model incorporating additive genetic (A), shared- (C), and non-shared (E) environmental effects, additive genetic factors (A) were found to account for 0% variation in resistome diversity at T1, 43% at T2 and 49% at T3. This trend was repeated at the ARG level. Additive genetic effects were found to significantly influence the abundance of *RlmA(ll)* and *patB* at T3 (Fig. 6a, Table S3), accounting for 32% and 52% of variation observed in these genes. Resistome diversity was also significantly shaped by non-shared (E) environmental factors (67% at T1 and 51% at T2 and T3).

We subsequently investigated specific shared (delivery mode, number of siblings and socio-economic status ([SES], based on

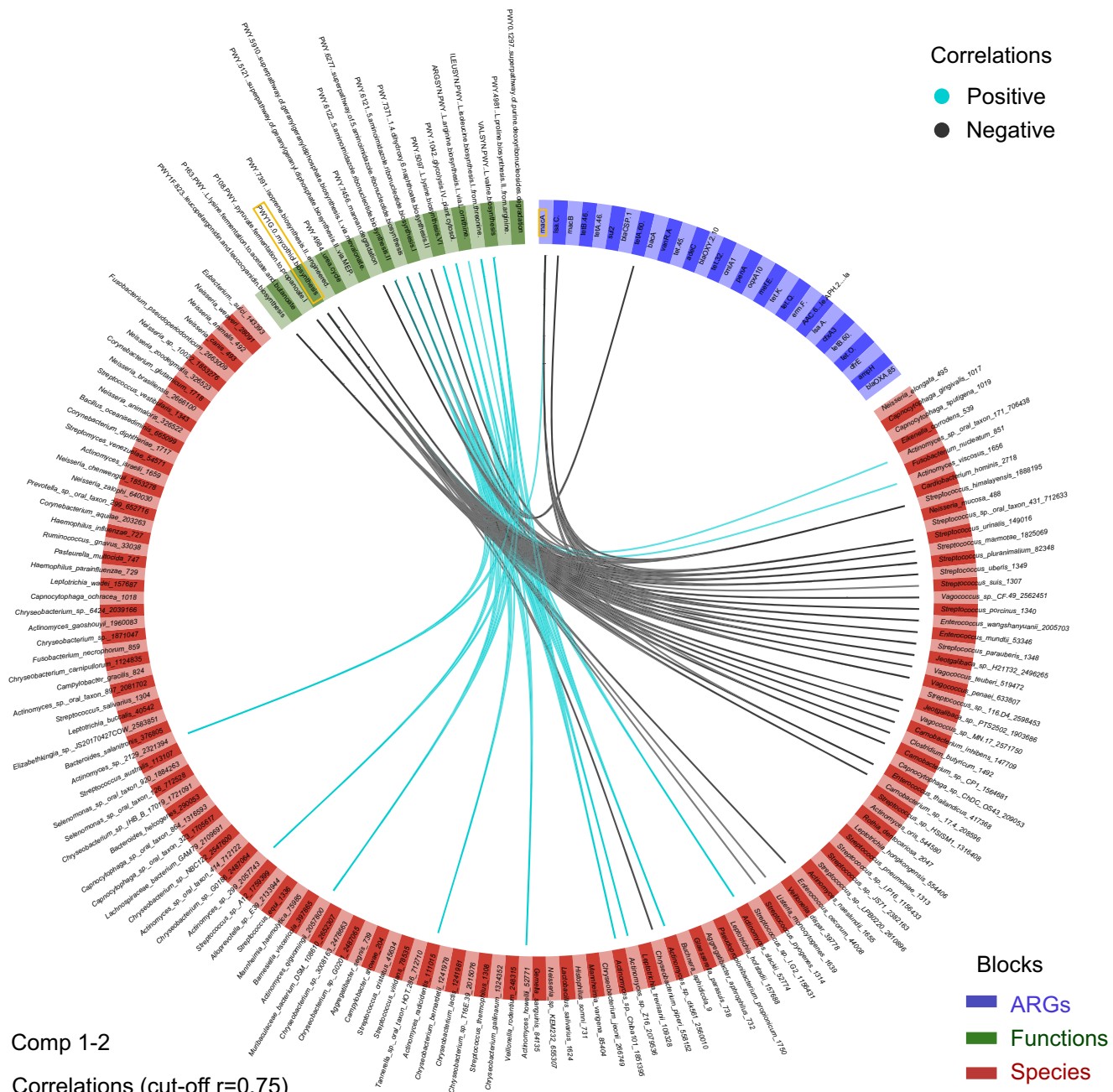

**Fig. 3 | Circos plot of correlations in the oral microbiome between AMR, species, and metabolic function, by dental development.** DIABLO was used to identify potential correlations between ARGs (TPM), species (CPM), and functional pathways (CPM). For each data type, DIABLO was performed on features above 0.01% abundance that had been TSS and CLR transformed. Displayed as a circos plot are correlations ≥0.75 between the different data types. Yellow boxes around text labels for *macA* and mycothiol biosynthesis indicate an ARG-function of interest correlation that contributes to clustering by time point/stage of dental development (Fig. S4).

postcode) and non-shared (early feeding practice, antibiotic exposure) environmental influences on the resistome phenotype (Supplementary Data 8 and 9) using linear mixed modelling. The resistome appeared to be influenced by dietary practices in early childhood. Significant associations (*p* values are Benjamini & Hochberg adjusted) were found at an ARG level with all core ARGs and early feeding practice at T2 with the abundance of *mefA* (*p* = 0.006), *msrD* (*p* = 0.014) and *RlmA(II)* (*p* = 0.014) increased in children exclusively breastfed until 6 months compared to those who were bottle fed (Supplementary Data 9). No significant associations were observed between delivery mode, antibiotic exposure, sibling numbers, or SES in the overall resistome diversity or specific ARGs.

Further investigation of the impact of antibiotic exposure was undertaken at a (i) population level by assessing if the composition of the resistome was associated with the national prescribing pattern and (ii) in a subset of the cohort (*n* = 132) at T3 to test the hypothesis that resistome composition is influenced by indirect antibiotic exposures like dietary protein. For the population level analysis, the relative abundance of specific classes of ARGs were compared to the defined daily dose per 1000 individuals (at each time point), using prescription data derived from the Resistance Map[25]. The overall relative AMR gene class abundance did not follow a linear relationship with Australian prescribing habits (Fig. 6b). There was a greater abundance of genes resistant to macrolides, compared to broad-spectrum penicillin

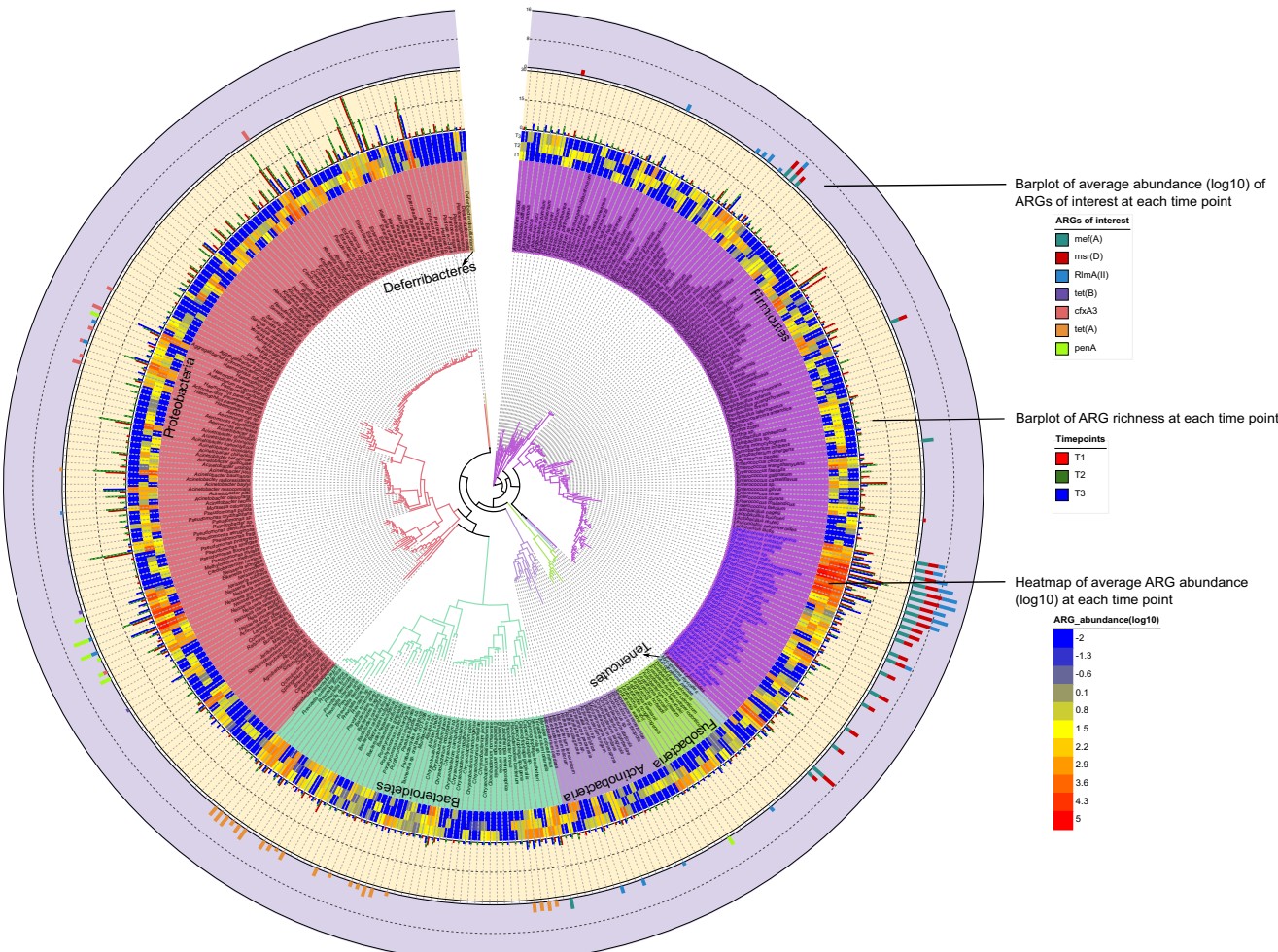

**Fig. 4 | Phylogenetic tree of ARG-carrying species.** A total of 356 ARG-carrying species across all time points were identified and used to reconstruct a species tree using 16 S rRNA sequences. The tree was generated using RAxML v8.0.0 and annotated using The Interactive Tree Of Life v6. Strains belonging to the same species were collapsed for abundance calculation. Species from the *Streptococcus* genus are highlighted in blue font. From inner to outer layer, presented per species, are: heatmap of average relative abundance (log10) of all ARGs (TPM normalised) detected per time point; bar plot showing number of ARG types (richness) per time point, and stacked bar chart showing the average relative abundance (log10) of seven ARGs (TPM) of interest including core ARGs (*mef(A)*, *msr(D),* and *RlmA(II)*) and those identified as being significantly associated with specific time points.

antibiotics that are more frequently prescribed. The relative abundance of genes resistant to broad-spectrum penicillins remained stable over time. Using the metadata collected, protein intake was defined as number of serves of lean meat, poultry, fish, and processed meat consumed per day based on a 7-day diet diary. This data was subsequently categorised based on the age-matched Australian guidelines[26]. No significant association was found between α-diversity, ARG abundances, and protein intake (Supplementary Data 9).

**Resistome composition is altered by the severity of caries and in response to restoration placement**

To assess if the resistome composition is altered in a diseased state (caries), we investigated if there were differences in overall abundance and diversity (α and β) between the caries free (CF, $n = 145$) and caries active (CA, $n = 66$) groups. There were no significant differences ($p > 0.05$) between groups for overall relative abundance or Shannon Index. Bray-Curtis scores (calculated using relative abundance) showed no clustering by group when plotted using non-metric multidimensional scaling. However, differences were observed at the ARG level. The CF resistome was richer than the CA resistome—83 ARGs compared with 57 ARGs, with 15 shared ARGs. The dominant AMR gene class in both groups, in terms of relative abundance, was macrolide

resistance (CF = 42% and CA = 44%), followed by β-lactams (CF = 19%, CA = 14%). The top ten most abundant genes were common to both groups, including *mef(A)* (a core ARG), which was the most abundant gene in both groups.

Caries status was defined using the International Caries Detection and Assessment System II[27] (ICDAS II). This is a two-digit coding system where the first digit denotes past caries experience (restorative status) and the second denotes current caries status. In this study the second digit was used to stratify the cohort by disease severity (healthy = 0, mild = 1–2, moderate = 3–4, severe = 5–6). To understand how the composition of the microbiome, resistome and insertion sequences changed over time in health and caries, we used the Jaccard index which measured the similarity in composition of samples based on caries status between different timepoints. A higher Jaccard score indicates greater similarity between groups. Of the four groups, the severe category had the highest compositional change in microbiome, resistome and insertion sequences (Fig. S7). In contrast, the healthy group showed highest similarity in both resistome and insertion sequences over time.

Differential abundance analysis was undertaken to assess if specific ARGs drove the resistome in the absence or presence of disease. Consistent with earlier findings, while no genes were identified as

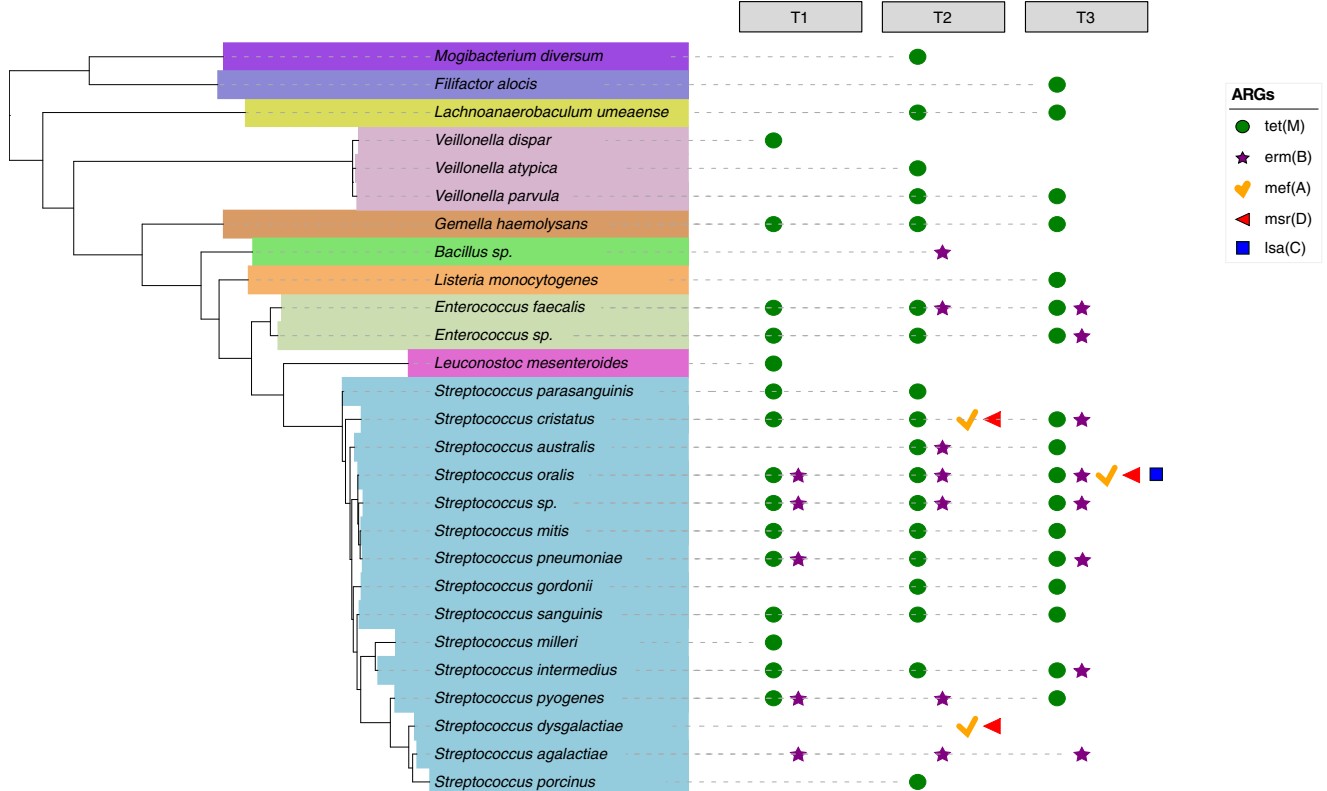

**Fig. 5 | Phylogenetic tree of oral species hosting both ARGs and *Tn916* transposase within 30 kb distance on the same contig at each time point.** The symbols show the specific ARGs found to be co-carried with *Tn916* per species. Multiple symbols indicate multiple ARGs were found at an individual contig level with *Tn916*. The tree was extracted from the tree shown in Fig. 4.

differentially abundant in either the CF or CA groups at a global level, (ANCOM-BC[28], MaAsLin2), 15 genes were identified as significantly abundant (ANCOM-BC) when comparing healthy resistomes with each disease category (Table S4). We also evaluated whether ARG abundance was altered based on the restorative status of the whole dentition to determine whether restoring caries had any impact on the resistome. Two ARGs, *blaOXA-85* and *tet(Q)* were significantly more abundant in children who had fillings, irrespective of their caries status (Table S4). *BlaOXA-85* was also identified as being significantly more abundant in health when compared to mild and moderate caries groups (Fig. 7a).

Finally, the most common ARG-carrying species in health and disease were *Streptococcal* spp. with *S. mitis* and *S. oralis* commonly associated with ARGs in both CF and CA groups. Genes that were significantly abundant in health (*aph6-Id*, *erm(f)*, *tet(W)*, *tetA(60)*) were associated with more bacterial genera than ARGs associated with disease (*blaOXA-85* and *tet(B)*) (Fig. 7b).

## Discussion

Our profile of 530 oral metagenomic samples from Australian twins over the first decade of life shows that the paediatric oral resistome is an inherent feature of the oral microbiome from infancy, which develops throughout childhood. While the known resistome accounts for <1% of the oral microbiome, AMR-associated mobile genetic elements (MGEs) were widespread across all time points including the known resistance carrier, the *Tn916* transposase family. Our novel finding indicates that this MGE is associated with an increasing number of ARGs across time points and is highly suggestive that the mobilisation potential of ARGs increase as children get older. We show that the composition of the oral resistome, like the oral microbiome, is influenced by host genetics and environmental exposures, particularly early feeding practices. Furthermore, the resistome composition was altered in dysbiosis based on the severity of dental caries and whether teeth were restored.

Our longitudinal metagenomic profiling of the resistome reveals it is both stable (persistent) and resilient (able to return to equilibrium after a temporary disturbance)[29]. Resistome resilience is evidenced by the fact that there were high levels of shared membership at an ARG level between time points (Fig. 2b) and across all time points. This is despite significant perturbations in the oral cavity in the first 2.5 years of life (tooth eruption, dietary changes, and growth). However, the composition of the resistome did exhibit significant temporal changes, with overall diversity increasing in the first two years of life before it stabilised at approximately age five (T3) (Fig. 2c). This contrasts with the gut resistome which undergoes a significant increase in ARG richness within the first 12 months of life[30,31].

Functional analysis revealed that the potential interactions between functional pathways and ARGs may aid the development of resistance. Later childhood (T3) was associated with increased abundance of *lsa(C)* and *macA* genes, which were positively correlated with an increased abundance of pathways for mycothiol biosynthesis. Mycothiol can detoxify antibiotics, enabling bacteria to survive antibiotic exposure, and hence develop resistance[32]. Furthermore, individuals with a higher diversity of ARGs were associated with a microbiome that had greater functional potential for biofilm building. Most of the biofilm building-related pathways (3/5) that increased in abundance in the higher resistome diversity group were related to sugar degradation (sucrose and galactose) and have previously been found to be enriched in caries[15]. However, transcriptomics is required to fully elucidate the relationship between AMR and bacterial metabolism.

To understand the dissemination of resistance via HGT we studied the taxonomic associations of the resistome. We found that the oral microbiome has the requisite constituents for a resistance "mobilome"

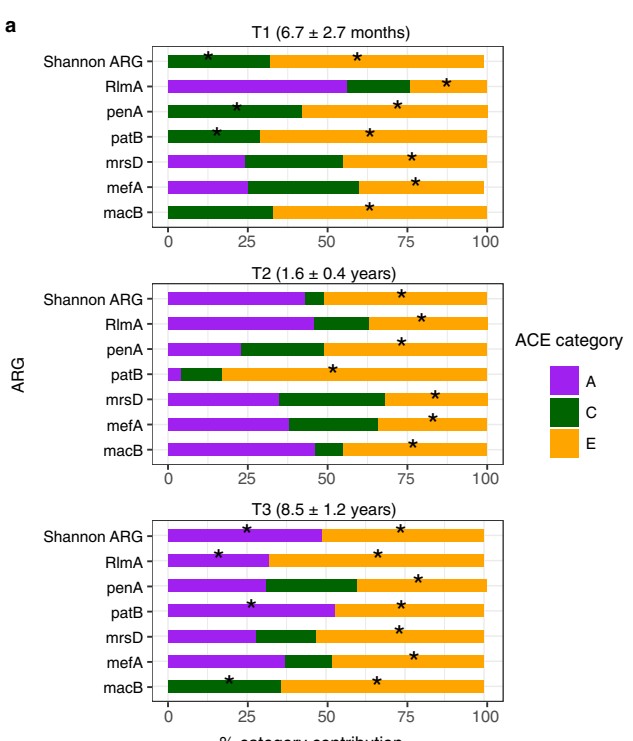

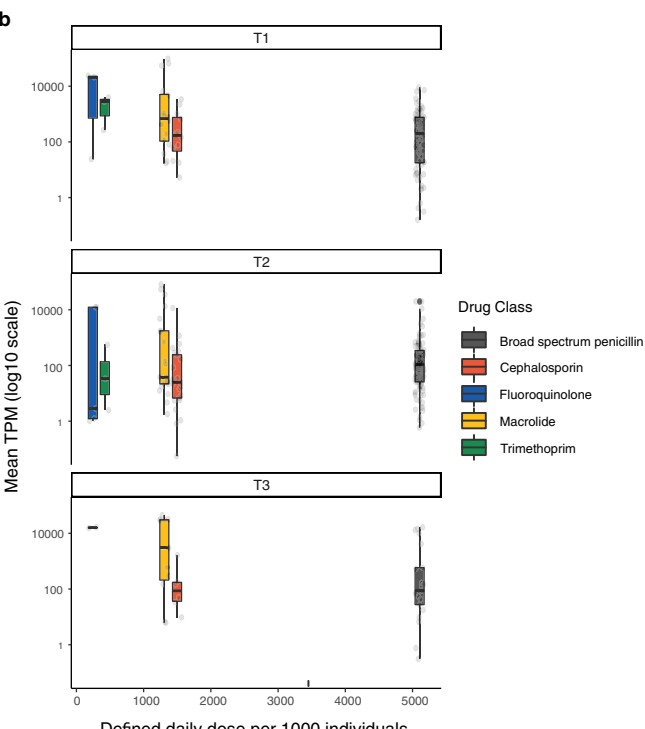

**Fig. 6 | Host-genetic and environmental influences on the resistome. a** Stacked bar chart shows standardised estimates of genetic effect/heritability (A), shared environmental effect (C), and individual environmental effects (E) per time point as percentages for composition metrics at a global (α-diversity) and specific ARG level. Estimates of A, C, and E were determined from a formal heritability analysis, using a variance components model approach for a quantitative trait, under the assumptions of the classic twin design. Black stars indicate significance ($p < 0.05$,

Table S3). **b** Boxplots of mean relative abundance of ARGs calculated as transcripts per million (TPM) for each AMR gene class against defined daily dose/1000 individuals from 2017 for Australia[69] per time point. Centre line of the boxplot represents the median, the minima and maxima of the box length corresponds to the first and third quartiles (the 25th and 75th percentiles), whiskers extend from the box to no further than 1.5*IQR (interquartile range) from the hinge. Data beyond the end of the whiskers are plotted individually.

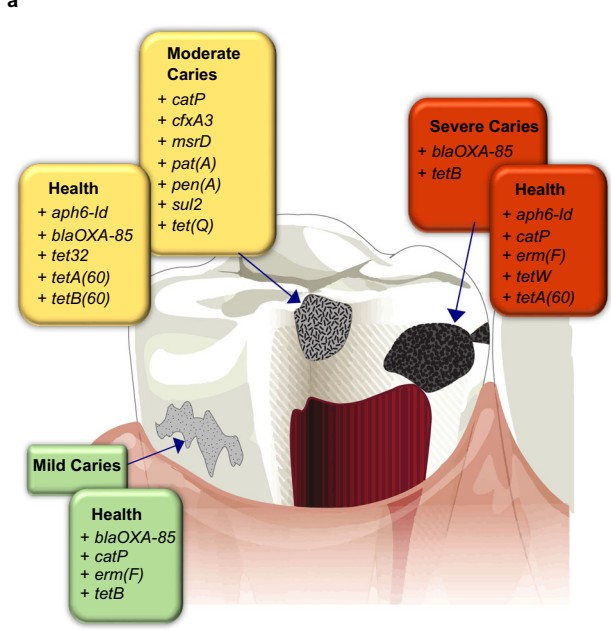

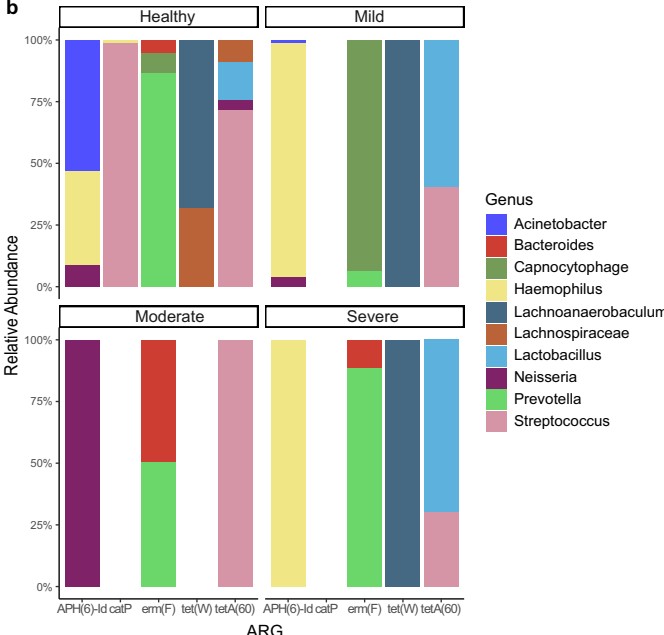

**Fig. 7 | The resistome in health and caries. a** Differential abundance analysis identified ARGs that were significantly abundant in health versus mild caries (green), health versus moderate caries (yellow), and health versus severe caries (red). **b** Depicts the top 10 bacterial genera associated with ARGs identified as

significantly associated with health and differing levels of caries severity. Caries severity was stratified by ICDAS II scores (health = 0, mild = 1–2, moderate = 3–4, severe = 5–6). 7a is adapted from Fig. 2 Simon-Soro and Miro 2014[70] with permission from Elsevier.

as we identified co-location of ARGs and IS in 27 species across the time points. While this finding reconfirms the increased HGT potential of biofilms[24] it also has systemic implications. Dental interventions like daily oral activities e.g., tooth brushing, may cause a bacteraemia[33]. In our population, *S. mitis, S. sanguinis,* and *S. anginosus* were identified as IS associated ARG carrying species. These oral commensals are known distant site pathogens causing infective endocarditis and ventilation-associated pnuemonia[34]. In systemic circulation these ARG/IS carrying species have the potential to cause infection which may be more resistant to treatment and transfer resistance in their new environment.

The *Tn916* family of transposases is extensively studied[7,24,35] and this AMR-associated IS is commonly associated with oral *Streptococcal* spp and known to facilitate the co-carriage of macrolide resistance gene *erm(B)* and tetracycline resistance gene *tet(m)*[24,36]. This was confirmed in our findings at T1. However, we were able to identify further associations at subsequent time points, specifically to the macrolide-resistant gene complex - *mef(A)/msr(D)* efflux pump. Notably this gene complex is highly abundant and prevalent in the respiratory resistome where it is also associated with Streptococci[37]. The presence of *mef(A)/msr(D)* carrying *Streptoccocal species* in the oral and respiratory resistome may be the result of anatomical proximity, but may also reflect the ability of the oral ARGs to mobilise to the respiratory tract via HGT.

As host genetics has been shown to influence oral microbiome composition[11], we used a twin study design to measure the relative impacts of the environment and host genetics on the ARGs present within these species. Our results indicate that host genetics potentially influences resistome diversity throughout the first decade of life. At T1, heritability was estimated as having no effect (A = 0%). We suspect this is an underestimate due to insufficient sample size at T1 ($n = 28$ twin pairs) compared to the later time points (T2 = 38 and T3 = 43 twin pairs). Under- or missing estimates of heritability due to underpowered sample sizes have been raised as a concern for twin studies assessing complex traits, such as the resistome, using genomic data[38]. Based on our findings, re-assessment of heritability of the oral resistome in children under the age of one year with a larger sample size would be required. Hence, the pattern of heritability could only be reliably determined for T2 and T3. The decline in heritability between T2 and T3, alongside an increase in both common and unique environmental effects on the oral resistome, would be expected given the greater exposure to diverse environments (e.g. dietary, antibiotics) of school aged children compared to infants[8]. While higher-powered studies are required, our results suggest that the absolute influence of the genes remains the same throughout childhood, however, their relative influence changes due to the changing contribution of the environment.

We analysed several direct (diet, antibiotic use) and indirect (family size, delivery mode, SES) environmental factors and were unable to identify many significant influences. Interestingly, our results imply that early feeding practices may lead to changes in resistome composition. Formula feeding is associated with increased AMR potential in the gut[39], however, we found an increase in abundance of core ARGs in breastfed babies. These ARGs were taxonomically associated with Streptococci, an early coloniser and commensal in the healthy oral cavity which may signify a protective function for these genes.

In our study, direct past antibiotic exposure did not alter resistome composition and this finding corroborated with the national prescribing data. Our results are possibly explained by the resilience of the oral microbiome to antibiotic exposure. The diversity of the resistome returns to baseline within 30 days of antibiotic exposure[40]. As children currently on or who had recently taken (3 months prior) antibiotics were excluded from analysis ($n = 2$), compositional changes due to antibiotic exposure cannot be detected in this study. However,

a recent gut resistome study found no correlation between ARG abundance and administration of antibiotics[41] suggesting that indirect exposures e.g., antibiotic use in food production, may influence resistome composition. While macrolides and tetracyclines are the most common antibiotics used for therapeutics and growth promotion in Australian piggeries and broiler farms[42], several other antibiotics are also used including fluroquinolones, aminoglycosides, and trimethoprim[43]. Our diet analysis of a subpopulation of children at T3 found that protein consumption does not influence resistome diversity. Future studies are required to investigate direct and indirect selective antibiotic pressures in the paediatric oral resistome.

Finally, we investigated the potential role of the resistome in health and caries given the widespread presence of ARGs in the oral microbiome. This was driven by the fact that microbial membership may not fully explain dysbiosis in caries[44–46] and microbial functionality, including AMR potential which may play a role in disease initiation and progression. As the composition of the attendant microbiome changes with disease progression[45], we hypothesised that this would be reflected in the resistome. Differential abundance analysis revealed *aph6-ld* and *tetA(60)* were consistently significantly more abundant in health compared to caries, and as severity of the disease progressed. The aminoglycoside gene *aph6-ld* is a known resident on MGEs and is capable of expression in a variety of bacterial species which may explain why it persists in the healthy resistome[47]. The β-lactam resistant gene *blaOXA-85* persisted as the disease progressed suggesting that this gene may be linked with caries-associated microbiota. In our cohort *blaOXA-85* was associated with *Fusobacterium* spp which has been previously implicated in caries[45].

As many current dental restorative materials are biologically inert, a focus of dental materials research is to create restorative materials which modulate the biofilm using antimicrobials to favour remineralisation of tooth structure[48,49]. We characterised the resistome in restored and unrestored dentitions to gain a better understanding of the resistome/restoration interface. *Tet(Q)* and *blaOXA-85* were significantly more abundant in resistomes where children had restorations. *BlaOXA-85* was generally associated with health, which suggests that the placement of a restoration favours a return to microbial homoeostasis. Our findings have direct translational impact for the development of future bioactive materials. The proposed antimicrobial actions of these materials need to transform the biofilm away from dysbiosis without inadvertently creating resistance. While we found that the known resistome accounts for less than 1% of the microbiome, the crucial factor is its ability to mobilise between species and disseminate to distant sites. Therefore, understanding how the resistome interacts with bacteria, tooth tissues, and restorative materials is critical to improving oral health outcomes.

In summary, our results confirm that ARGs are present in early life, forming a resilient and stable community within the oral microbiome with increasing potential to mobilise resistance as children get older. The intimate connections of oral cavity to the digestive tract, the respiratory and vascular system, as well as external perturbations, means mobilisation of the oral resistome has wide-ranging implications for systemic health and the spread of AMR. Our short read metagenomic investigation is limited by the inability to confirm possible associations between AMR, MGEs, bacterial species, and functional pathways. However, our conservative analysis reveals that surveillance of the oral resistome is essential, especially in early life. The first decade of life is a critical period in the development of the microbiome and its constituent communities which may have longer-term health impacts beyond childhood and the onset of tooth decay.

## Methods
### Study population
This study was undertaken with approval from the University of Adelaide Human Ethics Research Committee (H-2013-097 and H-78-2003).

Parents provided written informed consent for the use of samples and the data for this research.

Our cohort of 221 Australian children consisted of 108 twin pairs, 2 unpaired twins and one set of triplets identified from the 550 twin/triplet families enrolled in the Tooth Emergence and Oral Health study. Parents/caregivers were required to complete a series of questionnaires as part of the study. The sex of participants was assigned based on parental report. Sex-specific effects were not detected from analysis of ARG diversity or gene abundance. Hence, sex was not included in further analysis of the resistome. A standard medical history was taken at the clinical examination at T3. See Table S5, Supplementary Information for key population characteristics. At T3, the severity of caries was assessed using ICDAS II. In the 66 CA children, the mean ICDAS II score was $1 \pm 1.8$, while the remaining 145 were caries-free (ICDAS II score = 0).

From the 221 children, 542 oral biofilm samples were initially identified for genomic analysis, of which a total of 12 were excluded, making the final sample size 530. Two were excluded due to antibiotic use (within the past three months). Three were excluded due to extreme sequence depth variation from the mean. This included two with low sequence depth, T1657B_14042014_Q2_Q3 (0.339702 million target reads) and T1472B_25022007_D1_D2 (6.447160 million target reads), and one unusually high sequence depth sample, T1658A_6012009_D1_D2 (117.369 million target reads). We excluded seven samples that contained over 65% host DNA. The eligible samples were from 93 monozygous (MZ), 66 dizygous (DZ), and 59 opposite sex DZ (OSDZ) twins plus and one set of DZ/OSDZ/DZ triplets. One hundred and seventeen children (53%) were sampled at all three time points, 73 (35%) were sampled at 2 time points and 28 (12%) participants sampled at one time point only. While all twins/triplets were samples at the same time, not all individuals had a sample available that met the requirements for stage of dental development for all time points. For this reason, in addition to the post-sequencing removal of 12 samples, there was inconsistent sampling over time.

## Oral biofilm collection and storage

**Time points 1 and 2**. Parents/carers collected the initial samples from the twins by following detailed instructions given in Fig. S8. The samples were cultured within 7–10 days of reception to check for *Streptococcus mutans*. This was undertaken to check viability of the samples collected by the parents/carers.

Samples were plated on agar selective for *Streptococcus mutans* (TSYC20B) which was incubated for 72 h at 37 °C in an atmosphere of 95% nitrogen and 5% carbon dioxide. Following incubation *S mutans* was identified by characteristic colony morphology under a dissecting microscope. A sub-sample of positive colonies was confirmed by carbohydrate fermentation patterns. Each sample was assessed independently. After three contiguous positive cultures for *S mutans* it was assumed that *S mutans* had been acquired. The samples were then placed in black semi-solid transport media VMG II[50] and stored at 80 °C.

**Time point 3**. At the clinical visit, parents/carers and twins were instructed not to brush their teeth from 7 pm the previous evening until after the clinical visit and not to eat or drink in the half an hour before the appointment. Supra-gingival plaque biofilm samples were taken by a team of calibrated clinicians (registered dentist, oral health therapist or a supervised dental student) wearing sterile gloves. Oral biofilm samples were obtained using sterile Cultiplast® Tampone Swabs (LP Italiana, Milan, Italy). The labial/buccal surfaces and gingival margins of teeth in the maxillary and mandibular right-hand side were gently but thoroughly swabbed for 30 seconds per quadrant. The swab was inserted into a tube containing VMGII. The cotton tip was fully submerged, the wooden handle was broken so the swab was left in the media. The tube was sealed and placed directly on dry ice at collection. All samples were transferred to −80 °C freezer within 4 h of collection.

## DNA isolation and sequencing

DNA was extracted from 542 samples using DNeasy® PowerSoil®HTP 96 Kit (QIAGEN) as per Zhou et al.[51]. All samples were co-extracted with blanks to monitor for contamination. Nextera DNA Flex library preparation protocol was used to generate metagenomic libraries (n = 542 samples plus 12 duplicates) for sequencing using Illumina NovaSeq6000 (Illumina).

An average of 60 million, 134–150 bp, paired-end reads per sample were sequenced. Sequencing metrics are summarised in Supplementary Data 1. Validation of sequencing was assessed by calculating the interclass correlation coefficient (ICC) for sequencing metrics between duplicate samples (n = 12). There was good correlation between duplicate samples based on the number of filtered and assembled contigs (ICC = 0.86, $p = 6.1 \times 10^{-6}$) and level of host (human DNA) contamination (ICC = 1.0, $p = 3.1 \times 10^{-13}$). No issues could be identified with either sample collection (by calibrated clinicians) or extraction (completed by C.J.A and G.V.B), and host contamination was minimal (mean = 13% across all time points) with 68% of samples having <10% host contamination.

## Bioinformatic processing

We used our scalable Shotgun-Metagenomics-Analysis (v1.0)[52] to undertake bioinformatics processing. Workflow and scripts can be accessed here at https://github.com/Sydney-Informatics-Hub/Shotgun-Metagenomics-Analysis.

To remove host (human) contamination, reads were mapped against the hg38 human reference genome including ALT contigs (GRCh38) with BBtools 'BBmap' v 37.98[53]. Reads mapping to the human genome were discarded using the default BBMap settings, and the reads failing to map to GRCh38 were recorded in fastq format and taken forward as the 'target' reads for assembly, annotation, and statistical analysis.

Target (human/host-removed) reads were assembled with MEGAHIT v1.2.8[54]. Coverage of reads on assembled contigs was calculated with SAMtools 'coverage' tool[55], excluding reads/bases with mapping/base quality less than 20. Contigs with a mean mapping depth of coverage less than 1 were excluded from the assembly using seqtk v1.3-r113[56].

To assess the impact of conservative quality trimming and filtering (using FastQC v0.11.7[57] and MultiQC v1.7[58]) we assembled 10 randomly selected samples, which had host/human reads already removed. On these samples, we detected ARGs before and after performing quality filtering and trimming. We applied the following quality filtering and trimming parameters to the host-removed target reads using BBTools BBDuk: ktrim = r; k = 23; mink = 11; hdist = 1; tbo; qtrim = r; trimq = 8; ftm = 5; maq = 10. A greater number of ARGs were detected from assemblies produced from the unfiltered host-removed reads. Therefore, as adapter contamination was not detected and reads were of high quality, no further trimming or filtering was applied to the adapter and host removed reads (Fig. S9).

## Resistance and taxonomic profiling

Antimicrobial resistance genes were detected using ABRicate v0.9.9[59], using the following databases (Jan 2020): NCBI AMRFinder Plus, Resfinder, and Comprehensive Antibiotic Resistance Database (CARD). Each gene was filtered by coverage (≥ 70%) and identity (≥ 80%). Filtered ARGs were manually classified by type of antibiotic and resistance mechanism (Supplementary Data 2). To determine the abundance of ARGs in the microbiome, htseq-count v0.12.4[60] was used with a manually curated ARG list, in which one gene ID was selected for genes with multiple synonyms (Supplementary Data 10–12). Taxonomic profiling of contigs was performed with Kraken2 v.2.08[61] and Bracken v2.6.0[62] using contigs against the Kraken 'standard' database (build date 23 April 2020)[63]. Insertion sequences were identified from filtered contigs using Prokka v.1.14.5[64] and annotated using ISFinder

database[65] (update 2019-09-25). Functional profiling of contigs was performed using HUMAnN2 v2.8.2[66], with functional pathways in each sample classified against UniRef90 and chocophlan databases. Values reported are normalised as counts per million (CPM).

## Phylogenetic analysis

The 356 identified representative sequences from the ARG-carrying contigs were used to reconstruct a species tree using 16S rRNA sequences. The 16S rRNA sequences used to construct the species tree were retrieved from 16S NCBI refSeq database as of February 2022. Species not found in the 16S NCBI RefSeq database were obtained using NCBI genomes and blastn programme with $e$ value $1e^{-50}$. 16S alignments were generated using MAFFT[67] then trimmed with trimAl[68] under automated1. The 16S phylogenetic tree was generated using RAxML v8.0.0[69] under options (-m GTRGAMMER -f a -x 123 -N autoMRE -P 12345). Annotation of the 16S phylogeny was performed using the Interactive Tree Of Life v6[70], which incorporated the ARG-carrying species over time. Strains belonging to the same species were collapsed for abundance calculation.

## ARG abundance

For each sample, ARGs according to the curated ARG list were extracted from the ABRicate output and formatted as GFF. BAM files were created by first aligning the target reads back to the assemblies with BWA v0.7.17, sorting and indexing with SAMtools v1.10, and marking duplicate reads with GATK v4.1.5.0[71] using an optical pixel distance of 2500. The GFF and BAM files were used as input to htseq-count v0.12.4[60] with the following parameters: stranded no, min quality 20, type gene, mode union, non-unique none, secondary-alignments ignore and supplementary-alignments score. Transcript per million (TPM) normalisation was performed on the htseq-count output by dividing the read counts by the length of each gene in kilobases, which is the reads per kilobase (RPK) value. The RPK values in a sample are summed and then divided by 1,000,000. This is the per million scaling factor. TPM is calculated by dividing the RPK values by the scaling factor[72].

## Size of the resistome

To express the resistome (total number of ARGs) as a fraction of the microbiome (total number of genes), Prodigal v2.6.3 was used to predict protein coding sequences, per sample, from target contig assemblies obtained from MEGAHIT. DIAMOND's blastp tool v2.0.11 was used to query sample coding sequence (CDS) against the NCBI non-redundant (NR) database (prepared for DIAMOND with makedb). Prodigal CDS sequences were annotated as genes after applying cut-offs of ≥ 75% identity, ≥ 75% query coverage, 1E-6 evalue and an alignment length of at least 25[73]. Genes were classified as ARGs by position matching Prodigal CDS sequences and start or end gene positions to ABRicate ARG gene positions, curated in a previous step for non-redundancy.

## Statistical analysis

Scripts for resistome statistical analysis can be found at https://github.com/Sydney-Informatics-Hub/Shotgun-Metagenomics-Analysis. Alpha (Shannon Index), β-diversity (PCA, Bray Curtis distances), and Jaccard Index of the resistome were calculated using normalised relative abundance data in R (v4.0.2, https://www.R-project.org) with vegan v2.5-7[74]. Jaccard Index was calculated using the formula J(A, B) = |A∩B| / |A∪B| where A and B were feature compositions in different time points.

## Differential abundance analysis

Differential abundance analysis (DAA) was used to assess the impact of time point or stage of dental development and caries at T3 on the resistome. Prior to DAA, the data were filtered by removing low

abundance ARGs (cutoff: 0.01% counts) and ARGs present in only one sample. To assess the impact of time point, we used a linear mixed effects model (MaAslin2), where time point was treated as a fixed effect and individual as a random effect, to account for repeated sampling per person. Analysis was performed on ARGs present in at least 20% of the samples that had been normalised by total sum scaling (TSS) and log transformed. To ensure robust biological interpretation[75], we used two DAA methods (MaAslin2 v1.0.0 and ANCOM-BC v2.1.2), to investigate the impact of caries (mild, moderate, severe) on ARGs at T3. We then assessed if treating caries (restorations) impacted ARGs where caries status (caries free or affected) and restorative status (restored [yes] or unrestored [no]) were treated as fixed effects. Default parameters were used in both packages; however, alpha was adjusted to 0.25 in ANCOM-BC to be consistent with MaAsLin2.

## Temporal analysis of resistome composition

The relationship between resistome composition (Shannon ARG) as outcome and microbiome/species composition (Shannon SP) as predictor, cross-sectional and cross-time, was assessed using generalised estimating equation, which takes within twin pairs correlation into account. This estimation method does not down-weight the effect of outlier observations, if present, which may affect the estimated magnitude and significance of the relationship. Therefore, a scatter plot of outcome versus predictor was plotted (not shown), and outliers were identified by visual inspection and then excluded from subsequent analysis. This process was repeated to examine the relationship between Shannon ARG as outcome and functional pathways (Shannon function) as predictor. Results of the relationship were given in Supplementary Tables S2.1 and S2.2 respectively.

To examine the effect of changes over time between T2 and T1, T3 and T1, and T2 and T3 for global metrics, e.g., diversity and total relative abundance, the difference for each variable was calculated and then tested for normal distribution using skewness and kurtosis test for normality. A paired $t$ test was used as the distribution of the change score approximated normal. Results are presented in Supplementary Table S2.3.

## Correlation analysis

To examine the correlation among ARGs, species, and functional pathways and how these varied over time, we used DIABLO, which incorporated sparse Partial Least Squares Discriminant-Analysis for variable selection and network plotting for correlation visualisation with a cut-off 0.75. Input included three data blocks (ARGs, species, and functional pathways), which were filtered per block to remove low abundance features (below 0.01%) and normalised to account for sequence variation by Total Sum Scaling and Centre Log Ratio transformation. The grouping for the discriminant analysis component of DIABLO was set to time point. Correlations were reported between and within the data blocks if they reached a threshold of 0.75.

## Discriminant analysis

We used LEfSe v1.0.0 to investigate potential interactions between the abundance of 66 predicted functional pathways[76–80] involved in biofilm formation, in high- and low-resistome diversity groups, with a threshold of logarithmic LDA 2.0. We categorised samples according to their resistome diversity into the high and low Shannon score group. The cut off Shannon index scores were based on distribution, with a mean of 2.10 (below 2.10 = low diversity, above 2.10 = high diversity). Biofilm-related pathways were normalised on a per sample basis to sum to 1.0.

## Univariate twin analysis

Heritability was estimated for overall resistome diversity (Shannon) and individual ARG abundances at each time point using a classic twin design, assuming: twins shared the same environmental effects regardless of zygosity; MZ twins shared 100% of their genes; DZ twins

shared, on average, 50% of their genes. Under these assumptions, if there is a familial influence on a variable, the correlation within twin pairs will be greater than expected by chance; if the correlation is greater for MZ twins than for DZ twins, the presence of an additive genetic influence is supported; if the correlation within pairs is similar between MZ and DZ pairs, then a shared environmental influence is suggested. A formal heritability analysis was conducted using a variance components model approach for a quantitative trait[81], again under the assumptions of the classic twin design. This model assumes a multivariate normal distribution for a given variable across twin pairs, with the mean being a function of covariates such as age and gender and a residual variance. The variance is decomposed into an additive genetic variance, *a*, a common environmental variance, *c*, that is shared by twins within the same pair, and an individual environmental variance, *e*, specific to an individual, and these are assumed to be independent. These variances were then standardised with A (=$a/(a+c+e)$) representing the proportion of variance attributed to additive genetic factors (heritability), C (=$c/(a+c+e)$) representing the proportion of variance attributed to shared environmental factors, and E (=$e/(a+c+e)$) representing both the proportion of variance specific to individual environmental factors, and measurement error. The parameters in the model can be estimated by maximising the likelihood function, and the significance of estimated variance components can be tested using the likelihood-ratio test (e.g. ACE vs AE). In our application, however, only the full ACE model was fitted, using the package *mets*[82] in R (https://www.R-project.org), regardless of whether parameters were significant.

The influence of environmental factors on the resistome was investigated in STATA (http://www.stata.com) using a linear mixed model, accounting for correlation within twin pairs. All *p* values reported from the linear mixed models for specific ARGs (Supplementary Data 9 [9.2–9.4]) are corrected for multiple comparisons using the Benjamini & Hochberg method.

### Reporting summary

Further information on research design is available in the Nature Portfolio Reporting Summary linked to this article.

## Data availability

Sequence data that support the findings of this study have been deposited in European Nucleotide Archive with the project accession code PRJEB54673. The authors declare that all other data supporting the findings of this study (including source data for all figures) are available within the article, and provided in the supplementary information files. Databases used to analyse sequence data included: NCBI AMRFinder Plus (https://www.ncbi.nlm.nih.gov/pathogens/antimicrobial-resistance/AMRFinder/), Resfinder (https://cge.cbs.dtu.dk/services/ResFinder/), Comprehensive Antibiotic Resistance Database (CARD) (https://card.mcmaster.ca/), Kraken 'standard' database (https://ccb.jhu.edu/software/kraken/) and ISFinder database (https://isfinder.biotoul.fr/). Source data are provided in this paper.

## Code availability

All bioinformatics scripts that can be used to reproduce our analyses are available at https://github.com/Sydney-Informatics-Hub/Shotgun-Metagenomics-Analysis.

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

## Acknowledgements

The authors would like to thank the twins and their families for agreeing to participate in our study and for the support provided by Twin Research Australia and the Australian Multiple Birth Association. The authors acknowledge the technical support provided by the Sydney Informatics Hub and the University of Sydney's high-performance computing cluster, Artemis, for providing the computing resources that have contributed to the results reported herein. This work was also supported by computational resources provided by the Australian Government through National Computational Infrastructure, Gadi under the National Computational Merit Allocation Scheme and Intersect HPC Allocation Scheme. The authors wish to thank NCI support staff, in particular, Dr. Andrey Bliznyuk and Dr. Ben Menadue for their assistance with scaling the workflow. The authors thank Mara Cvejic (Institute of Dental Research, Westmead Centre for Oral Health) for creating Fig. 6b. This work is supported by the following grants from Intersect HPC Merit Allocation Scheme 2020 (C.J.A., S.S., C.E.W., and T.C.), Financial Markets Foundation for Children 2009–223 (T.E.H. and M.R.B), National Health & Medical Research Council APP349448 (T.E.H.), APP1006294 (T.E.H.) and APP1062911 (C.J.A and T.E.H.), Channel 7 Children's Research Foundation 161328 (T.E.H. and M.R.B.) and National Institute of Dental and Craniofacial Research 1R01-DE019665 (C.J.A, T.E.H., and M.R.B).

## Author contributions

S.S., C.A.S., F.E.M., C.J.A., and E.M. conceived the presented idea. M.R.B., T.E.H., S.S., G.V.B., T.E.H., and C.J.A. were involved in clinical examination and sample collection from the twins coordinated by M.R.B. and T.E.H. Metadata collection was collected and coordinated by M.R.B. and K.M.D. and curated by S.S. Laboratory work was undertaken by G.V.B. and C.J.A. R.S., C.E.W., S.S., E.M., T.C., H.W.L., and C.J.A. conceived the processing pipeline and C.E.W., T.C., and F.W. conducted the bioinformatics with support from E.M. S.S., C.A.S., F.W., and C.J.A. investigated and interpreted the results. Statistical analysis and interpretation were undertaken by Q.M.B., T.E.H., S.S., F.W., C.A.S., and C.J.A. C.J.A., F.W., S.S., and H.W.L. created visualisations. S.S. and C.J.A. co-wrote the paper with support from E.M., C.E.W., R.S., F.E.M., G.V.B., T.E.H., Q.M.B., and M.R.B. C.J.A. oversaw the project.

## Competing interests

The authors declare no competing interests.
