## [Peer Review File · Nature Communications]

REVIEWER COMMENTS

Reviewer #1 (Remarks to the Author):

The concept of the oral resistome, even in clinically healthy individuals who have not received antibiotics, is a very timely and important one to study. Some unclarities remain after review of this work:

1. The third aim of the study entails the investigation of the contribution of AMR to the presence and severity of dental caries, by comparing the resistome in caries free and caries affected children. It is nevertheless unclear why a population of patients with dental caries was selected, as this is a biofilm-caused disease for which a) antibiotic administration is not warranted, and b) there is no indication that ABR-microbes are the causative agent (it is rather a matter of oral species ecologically adapted to a carbohydrate-rich oral environment).
2. Further to the above point, one of the exclusion criteria of the study was recent antibiotic use. It can be argued that it is meritable to include patients who have undergone antibiotic treatment, as this would pose ABR pressure and deliver a more differences in ABR gene levels when comparing the two populations (antibiotic usage vs non-usage). This should be elaborated further in the discussion section.
3. It is also unclear what is the rationale for recruiting particularly twin individuals in this study. Genetic or environmental aspects related to twin studies do not seem to be adequately addressed within the study design or considered clearly in the discussion. The scientific value of the inclusion of twins in this study should be highlighted in the introduction and (where/if appropriate) in the discussion.

Reviewer #2 (Remarks to the Author):

In this manuscript, the authors aimed to characterize the development of the oral resistome in children across the first 10 years of life. They utilized a cohort of 221 twin or triplet children and aimed to sample at three timepoints, resulting in 535 oral metagenomes from before baby teeth (n=141), when participants had baby teeth only (n=180), and when participants had a mix of baby and adult teeth (n=214).

The authors utilized short read metagenomic sequencing, followed by annotation of the oral resistome. They report discriminatory ARGs by timepoint, along with their associated taxa. They also state that they found a significant increase in unique insertion sequences (ISs) between timepoints 1 and 3, with particular emphasis on Tn916. They report contigs carrying ARGs within 7kb of Tn916 for their potential for mobilization and find an abundance of Streptococcal species associated with Tn916. Utilizing the twin cohort, they apply a variance components model and state that they see the influence of genetic factors on resistome diversity and ARGs increasing across timepoints. To further test the impact of antibiotic exposure, the authors compared the resistome with average antibiotic prescription data in Australia, as well as protein intake (to account for indirect antibiotic exposure) and found associations between alpha diversity and protein intake. Finally, the authors examined how the oral resistome was altered in the presence of caries and found differences in abundance at the AMR gene class level, as well as AMR genes associated with varying degrees of caries severity.

This study utilizes a large cohort to investigate the development of the oral microbiome, as well as draw associations between specific taxa/ARGs and dental disease. The study design incorporates 3 longitudinal timepoints at well-defined stages of oral development. The authors do a good job of guiding readers through different categorizations in a clear manner (T1/T2/T3, CF/CA). They also utilize metadata effectively by incorporating them in their analyses of genetic vs environmental factors and diet logs.

However, there are some general concerns that need to be addressed. Overall the manuscript should be edited for style, as there are many run-on sentences. There is also a lot of redundant information between the main text, methods, and supplementary methods. Further, the authors do not report on rarefaction of the dataset (see below) or reasoning for inconsistent sampling (not all patients had all 3 samples). Further, there are additional analyses that could have been done but were not addressed, such as prediction of functional pathways (using HUMAnN3, for example) or microbiome changes over time. In the discussion, there is a lack of explanation on why they think genetic factors play a greater role later in life. Finally, the authors claim the mobilization of ARGs in the oral microbiome may affect other parts of the body, but this statement lacks supporting evidence.

Major Concerns:

1. The range of reads per sample (339k to 117 million) seems extremely wide (almost 3 orders of magnitude). Was there any rarefaction done to account for the effect of sequencing depth on species identification? What factors may contribute to this wide range of sequencing reads? Inconsistent sequencing depth could greatly impact all findings, especially diversity comparisons.

2. In Supplementary Methods (bioinformatics processing): What do the authors mean by forgoing quality filtering? Were adapter sequences kept, and human reads not filtered? If there was host contamination, that would explain the greater number of ARGs found from the raw reads. In the main methods, it says BBMap was used to filter human reads. Further clarification is needed.

3. Were there any twin pairs that did not have concordant sampling times? (i.e., did all twins submit exactly the same number of samples as their sibling?)

4. In figure 5, what patients were included in this analysis? Only patients that you have all 3 timepoints for? Also, why not use specific treatment of patients rather than daily dose per 1000 individuals, it seems like there could be a lot of variation at the individual level that's not reflected by the population averages.

5. Line 266: What does a high degree of stability mean? Should consider some sort of statistical analysis or get some metric for what normal oral microbiota variation is.

6. Line 307: where was the data on direct antibiotic exposure? Not mentioned previously in text but would have been much more interesting than the average dose per 1000 individuals.

7. Would like to see some analysis of functional pathways in these metagenomes as well, even if ARGs didn't necessarily change based on caries status I would expect change in genetic pathways.

8. Line135: how did the authors determine median 11,538bp to be "sufficient length" for correlation analysis?

9. Line153: how was 7kb determined as a good length for ARG and IS association?

Minor Concerns:

1. Line 36: One Health approach -> briefly define for reader.

2. Line 38: flip order of sentence -> "The oral microbiome comprises the second most significant... and is an important interface ... food. Therefore, characterizing the AMR potential of the human oral microbiome is important."

3. Line 46: “systemic development of antimicrobial resistant infections”
4. Line 70: “three aims:” (change comma to colon before starting a list)
5. Line 74: “design, and (iii) investigation”
6. The genes identified in line 91 should be called something other than ARGs if you are going to remove MDEPs from your ARG analysis.
7. Line 100: the last group should probably be ‘Other’ instead of antibiotics.
8. Line 108: tetracycline resistance genes
9. Line 109: “and this was joined by β -lactamase resistance genes (cfxA3 and penA).” Unclear what “joined by” means here. Also unclear where cfxA3 and penA are in Figure S3.
10. Line 117: should be plurality, not majority.
11. Line 119: remove comma after “core subpopulation”
12. Line 130: brief sentence explaining the analysis would be helpful.
13. Line 147: what test was used?
14. Line 163: what is meant by “core ARGs”? This term should be defined at initial use (also appears in Figure 2).
15. Line 169: would be easier to comprehend if the symbols A, E, and C were incorporated into main text
16. Line 178: Table S16 should be bumped up to S11 to reflect the order they appear in writing.
17. Line 189: is this prescription data from Resistance Map age-matched to the cohort?
18. Line 261-262: would make more sense if resilient and stable were switched.
19. Line 277: mobilome spelling
20. Line 286: “compositional similarity between the oral and respiratory resistome”. Is there any other supporting data besides the Streptococci? If not, should rephrase.
21. Line 294: typo with two parentheses “(E)”
22. Line 296: why does environmental variance increase again at T3?
23. Line 371-372: what was the total # patients disease severity was assessed for? Only have the number that were caries free, not caries affected.
24. Line 301: are social economic status and location used interchangeably in this study? Is SES defined by address?

25. Line 326: "alters with" -> "changes with" or "is altered with"

26. Line 338: "may impact on" -> "may impact"

Figure Comments

1. Supplementary Figures overall need more explanation in caption.
2. Figure 1 bottom panels lack an x-axis label. How were the bin widths decided? Should be consistent across timepoints, or just collapse the timepoints since no comparisons were made within timepoints.
3. Figure 2: What are core ARGs? Should define in text.
4. Figure 2b: were there no ARGs in common between T2 and T3 that weren't in T1? Consider including a 0 if so.
5. Figure 3 is difficult to read, highlighting the emphasized taxa *Streptococcus* spp in figure would help legibility.
6. Figure 4: are only these 4 genes ever associated with Tn916?
7. Figure 5: what is the unit for "defined daily dose/1000 individuals"? or is the unit "# doses"?
8. Figure S2 y-axis is Count(log10). It is unclear what this unit is and needs elaboration (read count? RPKM? TPM?)
9. Figure S3 x-axis is unclear and could be specified in caption.
10. Figure S4: what do the panel titles mean? i.e., T1023A? Are they genes? Samples?

Point by Point Response

Manuscript number: NCOMMS-21-16739A-Z

Title: Development of the oral resistome during the first decade of life

Please find below our point-by-point response to the reviewers' comments. The reviewers' comments are *italicised* and our responses are provided sequentially following reviewers' comments. Line numbers refer to the updated, track change version of the paper.

Reviewer #1 (Remarks to the Author):

The concept of the oral resistome, even in clinically healthy individuals who have not received antibiotics, is a very timely and important one to study. Some unclarities remain after review of this work:

1. The third aim of the study entails the investigation of the contribution of AMR to the presence and severity of dental caries, by comparing the resistome in caries free and caries affected children. It is nevertheless unclear why a population of patients with dental caries was selected, as this is a biofilm-caused disease for which a) antibiotic administration is not warranted, and b) there is no indication that ABR-microbes are the causative agent (it is rather a matter of oral species ecologically adapted to a carbohydrate-rich oral environment).

Reply: As indicated above, caries is a biofilm mediated disease which is not treated with antibiotics. Therefore, why does our study investigate a cohort of children with and without caries? There are two primary reasons, (A) the biological link of the oral biofilm, which mediates caries and antibiotic resistance, and (B) the potential impact of current and future caries treatments on antibiotic resistance. We have clarified both points in our revised manuscript, as detailed below.

- A. Oral biofilm: While caries is primarily about species change, we currently have no idea about the interplay of this oral microbiome, biofilm mediated disease and antibiotic resistance. Given we have shown the widespread presence of antimicrobial resistance in oral biofilm, it is of interest to investigate potential interaction between both species membership and biofilm related functional pathways involved in caries, and antibiotic resistance (updated Results, line numbers 240 – 250 and Discussion, line numbers 350 – 361).
- B. Caries treatment: While current treatment for caries does not include antibiotics, there is the issue that current and future treatments may cause antibiotic resistance. Current restorative materials have unknown effects on antibiotic resistance. In addition, newly developed antimicrobial approaches for caries treatment, such as the antimicrobial peptide, nisin produced from *Lactococcus*, are being assessed and have also been found to induce antimicrobial resistance in other oral species (updated Introduction, line numbers 66 – 69). Given this, it is important to understand the baseline, the resistance in caries versus not, and how it is modified by current treatment, as this may inform the design of new treatments to reduce potential resistance effects (updated Discussion, line numbers 358 – 361).

2. Further to the above point, one of the exclusion criteria of the study was recent antibiotic use. It can be argued that it is meritable to include patients who have undergone antibiotic treatment, as this would pose ABR pressure and deliver a more differences in ABR gene levels when comparing the two populations (antibiotic usage vs non-usage). This should be elaborated further in the discussion section.

Reply: Future studies are planned to assess the impact of antibiotics on the oral resistome, however, this was not testable with the current cohort because only two children were taking antibiotics at the time of sampling (updated Supplementary Methods, line number 8). Given this, we had an insufficient sample number in the current study to investigate the difference in resistome composition based on direct antibiotic use at time of sampling.

Despite this limitation, our study's use of a longitudinal, epidemiological survey design, can provide insights into the presence of antimicrobial resistance (AMR) at the population level in the oral microbiome of children, independent of direct antibiotic exposure.

We used a convenience sample from a well-documented longitudinal study with a genetically informed data structure (classical twin design). This study was not initially designed to assess AMR, however, it is a valuable population to use for metagenomic surveillance as appropriate sampling was undertaken and relevant metadata collected.

To address this point, we have indicated in the discussion the need for future interventional studies (updated line numbers 335 – 336).

3. It is also unclear what is the rationale for recruiting particularly twin individuals in this study. Genetic or environmental aspects related to twin studies do not seem to be adequately addressed within the study design or considered clearly in the discussion. The scientific value of the inclusion of twins in this study should be highlighted in the introduction and (where/if appropriate) in the discussion.

Reply: We agree this was not clearly articulated. The scientific value of a twin study design to measure the overall contribution of host genetic versus environmental factors to explaining variation observed in the oral resistome and how this potentially changes through different stages of childhood has been highlighted in the Introduction (updated line numbers line 53 – 59) and Discussion (updated line numbers 311 – 313).

Reviewer #2 (Remarks to the Author):

In this manuscript, the authors aimed to characterize the development of the oral resistome in children across the first 10 years of life. They utilized a cohort of 221 twin or triplet children and aimed to sample at three timepoints, resulting in 535 oral metagenomes from before baby teeth (n=141), when participants had baby teeth only (n=180), and when participants had a mix of baby and adult teeth (n=214).

The authors utilized short read metagenomic sequencing, followed by annotation of the oral resistome. They report discriminatory ARGs by timepoint, along with their associated taxa. They also state that they found a significant increase in unique insertion sequences (ISs) between timepoints 1 and 3, with particular emphasis on Tn916. They report contigs carrying ARGs within 7kb of Tn916 for their potential for mobilization and find an abundance of Streptococcal species associated with Tn916. Utilizing the twin cohort, they apply a variance components model and state that they see the influence of genetic factors on resistome diversity and ARGs increasing across timepoints. To further test the impact of antibiotic exposure, the authors compared the resistome with average antibiotic prescription data in Australia, as well as protein intake (to account for indirect antibiotic exposure) and found associations between alpha diversity and protein intake. Finally, the authors examined how the oral resistome was altered in the presence of caries and found differences in abundance at the AMR gene class level, as well as AMR genes associated with varying degrees of caries severity.

This study utilizes a large cohort to investigate the development of the oral microbiome, as well as draw associations between specific taxa/ARGs and dental disease. The study design incorporates 3 longitudinal timepoints at well-defined stages of oral development. The authors do a good job of guiding readers through different categorizations in a clear manner (T1/T2/T3, CF/CA). They also utilize metadata effectively by incorporating them in their analyses of genetic vs environmental factors and diet logs.

However, there are some general concerns that need to be addressed. Overall the manuscript should be edited for style, as there are many run-on sentences.

Reply: The manuscript has been extensively reviewed with a focus on using shorter sentences and an active voice. We have not highlighted these style changes, given the large number would have made other changes to reviewers' comments difficult to see.

There is also a lot of redundant information between the main text, methods, and supplementary methods.

Reply: Main text, methods and supplementary methods have been reviewed to eliminate redundancies. Methodology for additional analysis requested/done has been highlighted in yellow in Supplementary Information including the following sections - updated line 121 temporal analysis of the resistome, line 137 correlation analysis and 146 Discriminant analysis. Please note that a minimal amount of redundancy is required to have a level of detail in text.

Further, the authors do not report on rarefaction of the dataset (see below) or reasoning for inconsistent sampling (not all patients had all 3 samples).

Reply: Rarefaction of the dataset is addressed in response to Major Concern, Point 1, below.

We agree the reasoning for inconsistent sampling was not clearly described in the original manuscript. There are three reasons why not all samples were represented at each time point:

- A. Sample availability: Not all individuals had a sample available that met the requirements for stage of dental development (e.g., edentulous, deciduous and mixed).
- B. Sequence quality issues: For a minority of samples ($n=10$), twins had a sample excluded at the post-sequence quality filtering stage, due to sequence depth issues or high host contamination.
- C. Exclusion criteria: Two samples were excluded due to taking antibiotics at time of sampling.

Please note, for each time point, if a sample was taken, twin pairs were always sampled concurrently.

To clarify the reasons for inconsistent sampling, we have updated the Supplementary Methods, (updated line numbers 7 – 20, Supplementary file).

Further, there are additional analyses that could have been done but were not addressed, such as prediction of functional pathways (using HUMAnN3, for example) or microbiome changes over time.

Reply: We did look at species change in microbiome composition and diversity overtime, but given the focus of the paper is AMR, we did not want to distract from this by focusing on time change in species or function alone, as this would be a different aim/study.

In the original submission, we did not look at the relationship between potential changes in functional genes and AMR in the oral microbiome overtime. As such, we have included analysis examining the interaction between predicted functional pathways (HUMAnN2 output), species and ARG abundance, overtime, using a multi-block discriminant analysis approach (DIABLO). As would be expected, most correlations identified were between species and functional pathways. However, we did find 11 correlations between ARGs and functional pathways. Interestingly, we identified an increase in positive correlations overtime between macrolide resistance genes and functions that enabled the bacteria to detoxify antibiotics, e.g., survive antibiotic exposure, which may aid in the development of resistance. We have included this new analysis in the Results (updated line numbers 133 – 141), adding a new Figure (Figure 3) and described the implications in the Discussion (updated line numbers 276 – 281). Details of the HUMAnN2 and DIABLO analysis are included in the Methods (updated line numbers 421 – 424) and Supplementary Methods (updated line numbers 137 – 145).

In the discussion, there is a lack of explanation on why they think genetic factors play a greater role later in life.

Reply: We have outlined in the Discussion (updated line numbers 303 – 315) that changes in the proportional effect of host genetics over time, we think, is being largely driven by the changing impact of environment. To clarify this, we have revised this section, see updated line numbers 311 – 313.

Finally, the authors claim the mobilization of ARGs in the oral microbiome may affect other parts of the body, but this statement lacks supporting evidence.

Reply: We have updated the Discussion (line numbers 285 – 292) to provide a clearer description, highlighting which oral bacterial species we identified containing ARGs that are known to cause systemic infections

Major Concerns:

1. The range of reads per sample (339k to 117 million) seems extremely wide (almost 3 orders of magnitude). Was there any rarefaction done to account for the effect of sequencing depth on species identification? What factors may contribute to this wide range of sequencing reads? Inconsistent sequencing depth could greatly impact all findings, especially diversity comparisons.

Reply: Thank you for picking this up! We have reviewed sequence depth and have removed 3 sequence libraries from the original dataset.

We removed:

- A. Two low sequence depth samples: T1657B_14042014_Q2_Q3 (0.339702 million target reads) and T1472B_25022007_D1_D2 (6.447160 million target reads)
- B. One usually high sequence depth sample: T1658A_6012009_D1_D2 (117.369 million target reads)

The read range is now 11,215,924 to 89,678,704 pairs per sample, with a mean sequence depth of 53 million sequences per sample (updated line numbers 84 – 85).

To address this, we have added to the Supplementary Methods a section on sequence depth filtering (updated line numbers) and included a histogram (Figure S1) to clearly display the original sequence range.

Due to the removal of these three samples, all analyses in the paper were re-run and the updated data and resultant Figures provided.

2. In Supplementary Methods (bioinformatics processing): What do the authors mean by forgoing quality filtering? Were adapter sequences kept, and human reads not filtered? If there was host contamination, that would explain the greater number of ARGs found from the raw reads. In the main methods, it says BMap was used to filter human reads. Further clarification is needed.

Reply: We agree that the wording of the quality filtering section was unclear. In brief, quality filtering included, in the following order: adapter removal, assessment of sequence quality, mapping to human genome and removal of human mapped reads, and for non-human reads (e.g., target), assembly, annotation and analysis. We have clarified this in the Methods (line numbers 401 – 407).

We also agree that our testing of our quality filtering steps was difficult to follow. To assess the impact of adding additional quality filtering, namely trimming host removed reads, we ran a comparison. On a random selection of ten samples, on adapter and host (human) removed reads, we further trimmed the reads and compared the sequence quality to un-trimmed reads. We found that both the further trimmed and untrimmed had similar sequence quality and no adapter contamination. Hence, we decided to leave reads without further trimming. We have revised the description of this quality filtering check in the Supplementary Methods (updated line numbers 61 – 68) and included the FastQC scores (Figure S7).

3. Were there any twin pairs that did not have concordant sampling times? (i.e., did all twins submit exactly the same number of samples as their sibling?).

Reply: Yes, all twins were concordantly samples.

4. In figure 5, what patients were included in this analysis? Only patients that you have all 3 timepoints for? Also, why not use specific treatment of patients rather than daily dose per 1000 individuals, it

seems like there could be a lot of variation at the individual level that's not reflected by the population averages.

Reply: Figure 6a (previously 5a) includes only children with samples analysed at all three timepoints

Figure 6b (previously 5b) is a general population analysis looking at how abundance of specific AMR classes in the resistome correlates with Australian prescribing practice. This approach was necessary as we did not have accurate data about children's direct exposure (rely on parent recall at dental examination, see point 6 below). Figure 6b has been revised to show daily dose per 1000 individuals at each time point.

5. Line 266: What does a high degree of stability mean? Should consider some sort of statistical analysis or get some metric for what normal oral microbiota variation is.

Reply: This term was ambiguous and we have removed the sentence.

6. Line 307: where was the data on direct antibiotic exposure? Not mentioned previously in text but would have been much more interesting than the average dose per 1000 individuals.

Reply: We did analyse the impact of direct antibiotic exposure on the resistome using linear mixed modelling (see Results, line numbers 182 – 185). Data about direct antibiotic exposure was limited to yes/no and is outlined in Supplementary Table S11. This binary response is due to recall bias and minimal information about types of antibiotics prescribed. Given the lack of detailed metadata on past antibiotic intake for the cohort, we decided to additionally use a population-based approach - dose per 1000 individuals as a proxy for antibiotic intake.

This is a surveillance study, however future projects specifically designed as interventional studies are required to answer this question (updated line numbers 335 – 336).

7. Would like to see some analysis of functional pathways in these metagenomes as well, even if ARGs didn't necessarily change based on caries status I would expect change in genetic pathways.

Reply: Functional analysis has now been included which looks specifically at correlations between predicted functional pathways involved in biofilm function, caries status (based on ICDAS II scores) and α -diversity scores of the resistome (categorised as high and low). To prevent the paper from shifting focus to the topic of metabolic pathways in caries, we concentrated on pathways related to biofilm formation, due to the oral biofilms role in both caries and antibiotic resistance. Our hypothesis was that children with caries would have an enrichment of biofilm related pathways, and given this greater biofilm building ability, greater diversity of ARGs.

Our new findings are presented in the Results (updated line numbers 240 – 250) and evaluated in the Discussion (updated line numbers 350 – 357). While we identified a joint relationship between increased ARG diversity and increased abundance of pathways related to caries, such as sugar metabolism, this initial analysis makes clear that to answer the question about changes in functional pathways in disease and their relation to the resistome requires transcriptomics.

8. Line 135: how did the authors determine median 11,538bp to be "sufficient length" for correlation analysis?

Reply: A new reference, Durrant *et al* 2020¹ (updated line 143) has been added to provide the rationale for the contig length for correlation analysis (see Figure 1). Our rationale is based on the approximate length of an IS, which is 1500bp and gene, which is 800 bp. Therefore, a contig length of >3-4kbp is required to establish a good association.

9. Line 153: how was 7kb determined as a good length for ARG and IS association?

Reply: To provide justification for this distance, we closely inspected the literature, looking specifically at co-location of Tn916 and ARGs². This revealed the longest Tn916 gene is Tn6000 (33.2kb). Therefore, we changed the cutoff length to 30kb which is large enough to include all cases given the length of variation of Tn916 family. This provided more associations between this IS and ARGs, which has been updated in the Results (line numbers 158 – 160) and Table S8.

Minor Concerns:

1. Line 36: *One Health approach -> briefly define for reader.*

Reply: A brief explanation of One Health has been provided for the reader, updated line numbers 37 – 39.

2. Line 38: *flip order of sentence -> “The oral microbiome comprises the second most significant... and is an important interface ... food. Therefore, characterizing the AMR potential of the human oral microbiome is important.”*

Reply: Sentence has been flipped as advised, updated line numbers 39 – 42.

3. Line 46: *“systemic development of antimicrobial resistant infections”*

Reply: Updated line number 46: edited as advised.

4. Line 70: *“three aims:” (change comma to colon before starting a list)*

Reply: Updated line number 73: edited as advised.

5. Line 74: *“design, and (iii) investigation”*

Reply: Updated line numbers 77: edited as advised.

6. *The genes identified in line 91 should be called something other than ARGs if you are going to remove MDEPs from your ARG analysis.*

Reply: Updated line number 90: edited as advised.

7. Line 100: *the last group should probably be ‘Other’ instead of antibiotics.*

Reply: Deleted antibiotics but did not replace with other as updated line number 96-99 now reads: “The number of AMR gene classes decreased over time (20 at T1, 23 at T2 and 17 at T3) with five classes (macrolides, β -lactams, macrolide/lincosamide, fluoroquinolones and tetracycline) accounting for 90% of the total relative abundance across all time points.”

8. Line 108: *tetracycline resistance genes*

Reply: Updated line number 107: edited as advised.

9. Line 109: *“and this was joined by β -lactamase resistance genes (cfxA3 and penA).” Unclear what “joined by” means here. Also unclear where cfxA3 and penA are in Figure S3.*

Reply: This description and Figure S3 was unclear. We have now updated the results description (updated line numbers 103 – 112) using the vector loading data from the PCA to indicate which ARGs most contribute to the clustering observed on the PCA. We have replaced Figure S3 with a biplot of the PCA and vector loadings.

10. Line 117: *should be plurality, not majority.*

Reply: Updated line number 115: edited as advised.

11. Line 119: *remove comma after “core subpopulation”*

Reply: Updated line number 117: edited as advised.

12. Line 130: *brief sentence explaining the analysis would be helpful.*

Reply: We have updated the description of the diversity analysis (updated line numbers 128 – 130) and includes a brief description of the analysis undertaken: “We assessed whether the overall diversity of species and predicted functional pathways in the oral microbiome was predictive of the diversity of resistome, at each time point (Figure 2c) using generalised estimating equation (Supplementary Methods).”

13. Line 147: *what test was used?*

Reply: Update line 151-153 now reads “Insertion sequences were identified in all 530 metagenomes and the number of unique IS (richness) significantly increased between T1 and T3 (13.4% increase, $p < 0.0001$, paired t-test) (Table S4 & S8).”

14. Line 163: *what is meant by “core ARGs”? This term should be defined at initial use (also appears in Figure 2).*

Reply: The term “core” has been defined in update line numbers 118 – 119. The term core ARGs has been removed the legend for Figure 2.

15. Line 169: *would be easier to comprehend if the symbols A, E, and C were incorporated into main text*

Reply: Updated line number 175: edited as advised.

16. Line 178: *Table S16 should be bumped up to S11 to reflect the order they appear in writing.*

Reply: Table S16 has been renamed S14 to reflect the order it appears in the writing and is now included in the Supplementary Tables excel dataset.

17. Line 189: *is this prescription data from Resistance Map age-matched to the cohort?*

Reply: The data from the Resistance Map represents data from all ages including children. As advised in response to Major Concerns, point 6, this was a population-based approach informed by analysis done in Carr et al.³

18. Line 261-262: *would make more sense if resilient and stable were switched.*

Reply: Updated line numbers 268 – 269: edited as advised.

19. Line 277: *mobilome spelling*

Reply: Updated line number 284: edited as advised.

20. Line 286: *“compositional similarity between the oral and respiratory resistome”. Is there any other supporting data besides the Streptococci? If not, should rephrase.*

Reply: This has been rephrased and reads “The presence of *mef(A)/msr(D)* carrying Streptococcal species” (updated line numbers 299 – 300).

21. Line 294: *typo with two parentheses “(E)”*

Reply: Updated line number 310: edited as advised.

22. Line 296: *why does environmental variance increase again at T3?*

Reply: Please see updated line numbers 311 – 313, “At T3, environmental variance increases again as the children have greater exposure to diverse environments.”

23. Line 371-372: *what was the total # patients disease severity was assessed for? Only have the number that were caries free, not caries affected.*

Reply: The total number of children with caries have now been included. Updated line numbers 388 – 389, “At T3, the severity of caries was assessed using ICDAS II26. In the 66 CA children, the mean ICDAS II score was 1 ± 1.8 , while the remaining 145 were caries free (ICDAS II score = 0).”

24. Line 301: *are social economic status and location used interchangeably in this study? Is SES defined by address?*

Reply: Social economic status was assessed using postcode and this has been included in the main text (updated line 183) The process of categorisation is detailed in the legend in Supplementary Table S12.

25. Line 326: *“alters with” -> “changes with” or “is altered with”*

Reply: Updated line number 341: edited as advised.

26. Line 338: “may impact on” -> “may impact”

Reply: Deleted, now reads: “We characterised the resistome in restored and unrestored dentitions to gain a better understanding of the resistome/restoration interface.” (updated lines 360 – 361).

Figure Comments

1. *Supplementary Figures overall need more explanation in caption.*

Reply: All supplementary figure legends have been expanded.

2. *Figure 1 bottom panels lack an x-axis label. How were the bin widths decided? Should be consistent across timepoints, or just collapse the timepoints since no comparisons were made within timepoints.*

Reply: X-axis label has been added to the bottom panel as advised. The column is the same across all time points. However, it appears inconsistent as T3 has greater variation in age compared to T1 and T2. We chose not to collapse the time points as we believe the current figure provides a summary of the temporal changes observed and we have amended the results to discuss this, see updated line numbers 96 – 101.

3. *Figure 2: What are core ARGs? Should define in text.*

Reply: Addressed in response to Minor Concerns, point 14.

4. *Figure 2b: were there no ARGs in common between T2 and T3 that weren't in T1? Consider including a 0 if so.*

Reply: Figure 2b has been updated due to removal of samples. Eight ARGs are common between T2 and T3.

5. *Figure 3 is difficult to read, highlighting the emphasized taxa Streptococcus spp in figure would help legibility.*

Reply: To address this, we have highlighted Streptococcus spp in blue text.

6. *Figure 4: are only these 4 genes ever associated with Tn916?*

Reply: With the expanded distance, we now have five genes associated with Tn916. These are the genes expected based on previous work.

7. *Figure 5: what is the unit for “defined daily dose/1000 individuals”? or is the unit “# doses”?*

Reply: The unit for defined daily dose/1000 individuals is DDD/1000 individuals. However, the x-axis label in Figure 5b does not use the unit so figure can be readily interpreted by the reader

8. *Figure S2 y-axis is Count(log10). It is unclear what this unit is and needs elaboration (read count? RPKM? TPM?)*

Reply: Figure S2 has been deleted as this information was not discussed in the paper.

9. *Figure S3 x-axis is unclear and could be specified in caption.*

Reply: Figure has been replaced with a biplot PCA and caption updated.

10. *Figure S4: what do the panel titles mean? i.e., T1023A? Are they genes? Samples?*

Reply: Legend for Figure S4 now reads “The distribution length of contigs carrying ARGs was assessed from five randomly selected samples (T1028A, T1594B, T1121B, T1667A and T1023A)”

References

1. Durrant MG, Li MM, Siranosian BA, Montgomery SB, Bhatt AS. A Bioinformatic Analysis of Integrative Mobile Genetic Elements Highlights Their Role in Bacterial Adaptation. *Cell Host Microbe* **27**, 140-153.e149 (2020).
2. Roberts AP, Mullany P. Tn916-like genetic elements: a diverse group of modular mobile elements conferring antibiotic resistance. *FEMS Microbiol Rev* **35**, 856-871 (2011).
3. Carr VR, *et al.* Abundance and diversity of resistomes differ between healthy human oral cavities and gut. *Nat Commun* **11**, 693 (2020).

REVIEWER COMMENTS

Reviewer #1 (Remarks to the Author):

The authors have addressed efficiently the comments.

Reviewer #2 (Remarks to the Author):

In this rebuttal, the authors address all concerns brought up in our previous review. The adjustments made to help with redundancies in the text and sentence structure were very helpful and sufficient. We also appreciate the authors undertaking the effort to examine the changes in functional pathways over time and are excited to see that the authors found correlations between macrolide resistance genes and antibiotic detoxification. However, the new figure displaying these findings is difficult to read and some adjustments could help with visibility. Further, the authors claim to have found significant correlations between early feeding patterns and resistome phenotype. The p-values used in this analysis need to be corrected for multiple comparisons.

While we appreciate the additional analysis and structure the authors have provided for this publication, there are still a number of concerns that need to be addressed.

Major concerns

1. 309-313: Why would the environment stabilize at T2 but become more variable again at T3? There is no explanation as to why this would occur.
2. 182-185: The p-values mentioned are a few values from a large number of comparisons done and are not adjusted for multiple hypothesis testing. These corrections need to be made before they can be deemed significant.
3. line 240 onwards: The association between caries severity and biofilm related pathways remains unclear. Either remove this mention of "caries" from the hypothesis or elaborate on caries-associated findings. It is also seemingly contradictory to expect more severe caries to have more diverse resistomes, in light of the Result that CF resistomes were higher in richness than CA.

Minor concerns:

1. Line 134: "and ARGs, overtime, we used" -> "and ARGs over time, we used"
2. 142-144: The resistome had the most interactions with species as opposed to what? What do the numbers 44 and 55 signify?
3. 150: spell out insertion sequences (IS) at first instance
4. Figure 3 is difficult to interpret. Something needs to be done to make the lines differentiable, either removing the species section of the table, or adjusting thickness by correlation, etc. Some of the text labels are cut off along the top as well. "Expression" levels are also not clearly legible (what is the unit/axis?). may benefit from removing that layer of data from plot.
5. Figure 4: "Bacteroidetes" text is flipped.
6. 287: Maybe change to say that brushing/extraction can cause bacteremia, rather than does cause. This line also needs a citation.
7. 289: "causing , infective endocarditis, and" -> "causing infective endocarditis and ventilation"
8. 308-309: is there any previous literature discussing heritability being underestimated at earlier timepoints?
9. Table S4: make sure "sex" and "gender" are used distinctively

Point by Point Response

Manuscript number: NCOMMS-21-16739B

Title: Development of the oral resistome during the first decade of life

Please find below our point-by-point response to the reviewers' comments. The reviewers' comments are *Italicised* and our responses are provided sequentially following reviewers' comments. Line numbers refer to the updated, track change version of the paper.

Reviewer #1 (Remarks to the Author):

The authors have addressed efficiently the comments.

Reply: Thank you.

Reviewer #2 (Remarks to the Author):

In this rebuttal, the authors address all concerns brought up in our previous review. The adjustments made to help with redundancies in the text and sentence structure were very helpful and sufficient. We also appreciate the authors undertaking the effort to examine the changes in functional pathways over time and are excited to see that the authors found correlations between macrolide resistance genes and antibiotic detoxification. However, the new figure displaying these findings is difficult to read and some adjustments could help with visibility.

Reply: We agree and have updated Figure 3. Please see our detailed response below, under Minor concerns, point 4.

Further, the authors claim to have found significant correlations between early feeding patterns and resistome phenotype. The p-values used in this analysis need to be corrected for multiple comparisons.

Reply: Thank you for picking this up. We have now performed multiple test correction and the findings have remained the same. Please see our detailed response, under Major concerns, point 2.

While we appreciate the additional analysis and structure the authors have provided for this publication, there are still a number of concerns that need to be addressed.

Major concerns

1. 309-313: Why would the environment stabilize at T2 but become more variable again at T3? There is no explanation as to why this would occur.

Reply: We agree that the reasoning for the change in heritability over time was not clearly justified. In brief, we suspect that we may have 'missed' the effect of heritability at Time point 1 (T1) due to insufficient sample size and hence power. Heritability was calculated on 28 twin pairs at T1 compared to 38 at T2 and 43 at T3. To reliably estimate heritability of complex phenotypes, such as the resistome, generally requires large numbers of twin pairs. For example, in a study of the effect of heritability on the gut microbiome using metagenomic data, 127 adult twin pairs were assessed¹. Underestimates of heritability due to insufficient sample size have been raised as a concern for twin studies using genomic data².

In our study, we suspect the level of heritability at T1 is higher, potentially more at the level of T2 and then declined between T2 and T3. The lower level of additive genetic effects and increased environmental influence observed at T3 (~8 years) compared to T2 (1 – 2 years) is potentially due to increasingly diverse environmental exposures of school aged compared to younger children. To clarify this issue, we have re-worded the Discussion (updated line numbers 312 – 322), explaining that while we can describe the pattern of heritability from T2 onwards, we have concerns about the T1 additive genetic effect values due to potentially insufficient sample size.

2. 182-185: *The p-values mentioned are a few values from a large number of comparisons done and are not adjusted for multiple hypothesis testing. These corrections need to be made before they can be deemed significant.*

Reply: For the assessment of environmental factor influences (feeding, SES, antibiotics, etc.) on the resistome we have now performed multiple test correction using the Benjamini & Hochberg method (updated line numbers 197 – 200 and line numbers 446 – 448). Please note, we applied the p-value adjustment method to specific ARGs tested and not to the overall ARG diversity metric, given this is a single metric and not the same data type as the abundance of specific ARGs. Table S12.2, S12.3 and S12.4 now report the multiple test corrected p-values. Following multiple test correction, a significant relationship was still found at T2 between the abundance of ARGs and breast compared to bottle feeding.

3. *line 240 onwards: The association between caries severity and biofilm related pathways remains unclear. Either remove this mention of “caries” from the hypothesis or elaborate on caries-associated findings. It is also seemingly contradictory to expect more severe caries to have more diverse resistomes, in light of the Result that CF resistomes were higher in richness than CA.*

Reply: We agree that the relationship between caries, biofilm pathways and the resistome was unclear. As suggested, we have removed mention of caries from the hypothesis, focusing on the relationship between diversity of ARGs and abundance of biofilm promoting pathways. The updated description of these results (line numbers 142 – 152) and their discussion (line numbers 282 – 288) has been moved from the caries part of the paper to the section on the relationship between functional pathways and the resistome.

Minor concerns:

1. *Line 134: “and ARGs, overtime, we used” -> “and ARGs over time, we used”*

Reply: Updated line number 134: edited as advised.

2. *142-144: The resistome had the most interactions with species as opposed to what? What do the numbers 44 and 55 signify?*

Reply: This sentence has been revised and now reads “The DIABLO correlation analysis revealed that overall the resistome had more potential interactions with species than functional pathways.” (Updated line numbers 153 – 154).

3. *150: spell out insertion sequences (IS) at first instance*

Reply: Updated line numbers 161 – 162: edited as advised.

4. *Figure 3 is difficult to interpret. Something needs to be done to make the lines differentiable, either removing the species section of the table, or adjusting thickness by correlation, etc. Some of the text labels are cut off along the top as well. “Expression” levels are also not clearly legible (what is the unit/axis?). may benefit from removing that layer of data from plot.*

Reply: We agree that Figure 3 was too complicated. To improve visibility, we have made the following changes:

- A. Increased correlation cut-off from 0.7 to 0.75: This has reduced the number of correlations, while preserving the pattern from the original plot. Correlation values from the DIABLO analysis provided in Table S5 have been updated to only include those over 0.75.
- B. Changed connecting lines colour: To improve clarity, positive correlation values are now blue and negative black, making it easier to differentiate lines.
- C. Highlighted ARG-function correlation of interest: We have placed yellow boxes around the text labels for MacA and mycothiol biosynthesis.
- D. Removed expression data layer: In short, the expression level information was the vector loading values from the discriminatory analysis performed as part of DIABLO per data block. This showed which ARGs, species and functional pathways most contributed to the clustering by time point. We have presented this information as vector/plot loadings in the supplementary material (updated Figure S4).

E. Text labels are all in full, with no text cut-off.

5. Figure 4: “Bacteroidetes” text is flipped.

Reply: Figure 4 has been edited as advised.

6. 287: Maybe change to say that brushing/extraction can cause bacteremia, rather than does cause. This line also needs a citation.

Reply: Edited as advised with a new citation³. Updated line number 293 – 294 now reads “Dental interventions like daily oral activities e.g., tooth brushing, may cause bacteraemia.”

7. 289: “causing , infective endocarditis, and” -> “causing infective endocarditis and ventilation”

Reply: Edited as advised, comma has been removed in line 296.

8. 308-309: is there any previous literature discussing heritability being underestimated at earlier timepoints?

Reply: Please see response to Major concerns, point 1.

9. Table S4: make sure “sex” and “gender” are used distinctively

Reply: Title of Table S4 have been edited and now reads:

Table S4.1: Relationship between resistome composition (Shannon ARG) and microbiome/species composition (Shannon SP). Age and sex were not adjusted for because neither were found to have a significant effect.

Table S4.2: Relationship between resistome composition (Shannon ARG) and functional pathways (Shannon Function). Age and sex were not adjusted for because neither were found to have a significant effect.

References

1. Xie H, *et al.* Shotgun metagenomics of 250 adult twins reveals genetic and environmental impacts on the gut microbiome. *Cell systems* **3**, 572-584. e573 (2016).
2. Mayhew AJ, Meyre D. Assessing the Heritability of Complex Traits in Humans: Methodological Challenges and Opportunities. *Curr Genomics* **18**, 332-340 (2017).
3. Tomás I, Diz P, Tobías A, Scully C, Donos N. Periodontal health status and bacteraemia from daily oral activities: systematic review/meta-analysis. *J Clin Periodontol* **39**, 213-228 (2012).